# An anchoring complex recruits katanin for microtubule severing at the plant cortical nucleation sites

Noriyoshi Yagi [1,2,3], Takehide Kato [1], Sachihiro Matsunaga [2,4], David W. Ehrhardt [5,6], Masayoshi Nakamura [3✉] & Takashi Hashimoto[1✉]

Microtubules are severed by katanin at distinct cellular locations to facilitate reorientation or amplification of dynamic microtubule arrays, but katanin targeting mechanisms are poorly understood. Here we show that a centrosomal microtubule-anchoring complex is used to recruit katanin in acentrosomal plant cells. The conserved protein complex of Msd1 (also known as SSX2IP) and Wdr8 is localized at microtubule nucleation sites along the microtubule lattice in interphase Arabidopsis cells. Katanin is recruited to these sites for efficient release of newly formed daughter microtubules. Our cell biological and genetic studies demonstrate that Msd1-Wdr8 acts as a specific katanin recruitment factor to cortical nucleation sites (but not to microtubule crossover sites) and stabilizes the association of daughter microtubule minus ends to their nucleation sites until they become severed by katanin. Molecular coupling of sequential anchoring and severing events by the evolutionarily conserved complex renders microtubule release under tight control of katanin activity.

[1] Division of Biological Science, Nara Institute of Science and Technology, Nara, Japan. [2] Faculty of Science and Technology, Department of Applied Biological Science, Tokyo University of Science, Chiba, Japan. [3] Institute of Transformative Bio-Molecules (WPI-ITbM), Nagoya University, Nagoya, Japan. [4] Department of Integrated Biosciences, Graduate School of Frontier Sciences, University of Tokyo, Chiba, Japan. [5] Department of Biology, Stanford University, Stanford, CA, USA. [6] Department of Plant Biology, Carnegie Institution for Science, Stanford, CA, USA. ✉email: mnakamu@itbm.nagoya-u.ac.jp; hasimoto@bs.naist.jp

Microtubules in eukaryotic cells adopt various organizations, such as radial arrays centered on centrosomes in many animal cell types and transverse cortical arrays seen in anisotropically elongating and non-centrosomal plant cells[1]. Newly formed microtubules are either anchored at or released from the cellular nucleation sites and the control of such anchoring-release contributes to the array organization[2,3]. Mitotic spindle disanchored 1 (Msd1) was initially discovered as a yeast coiled-coil protein that is required to anchor the minus end of spindle microtubules to the spindle-pole body, the centrosome equivalent in fission yeast[4]. The synovial sarcoma X breakpoint protein (SSX2IP), an animal ortholog of Msd1, is localized to centriolar satellites that surround the centrosome and is essential for microtubule anchoring and centrosome integrity[5–7]. Msd1/SSX2IP is targeted to the microtubule nucleation sites partly by its interactions with components of the γ-tubulin ring complex (γTuRC)[4,5] and forms a functional protein complex with a WD40-repeat protein, Wdr8[8–11]. Cellular functions of Msd1 and Wdr8 in plant cells, which lack centrosomes, are not known.

In plant interphase cells, nascent microtubules are predominantly nucleated from lattice-bound γTuRC complexes on preexisting cortical microtubules, either as a dominant branch-forming pattern with a branch angle of ~40° or as a less frequent bundle-forming parallel nucleation manner[12–14]. The katanin complex, composed of the p60 AAA ATPase catalytic subunit and the p80 regulatory subunit[2], is required to sever the minus end of daughter microtubules at the branch junction sites and release them from the mother microtubules to generate free microtubules that migrate on the cell cortex by polymer treadmilling[14,15]. Katanin also severs microtubules at intersections where they cross over each other on the plant cell cortex[16,17], a process that promotes amplification and re-ordering of the cortical arrays in response to blue light stimulus[18]. Although the p80 regulatory subunit of katanin is required to target the p60 catalytic subunit to subcellular locations[19], it is not known whether specific factors differentially recruit katanin to these distinct subcellular sites in plant cells. During plant cell division, branched microtubules are nucleated along the lattice of preexisting microtubules in the plant cytokinetic phragmoplast array[20]. Katanin mutations affect organization and positioning of the spindle and the phragmoplast[21,22] but severing events are yet to be analyzed due to the high densities of mitotic microtubules. In this study, we show that the plant Msd1–Wdr8 complex is used not only to stabilize nucleation sites on cortical microtubule arrays but also to subsequently recruit katanin to these subcellular locations to sever daughter microtubules.

## Results

**The Msd1–Wdr8 complex is localized at cortical nucleation sites in *Arabidopsis* cells.** We first examined whether Msd1 and Wdr8 of *Arabidopsis thaliana* form heteromeric complexes as reported in non-plant organisms. In *Arabidopsis*, two *Msd1* orthologs (*Msd1a* and *Msd1b*) encode proteins with 75% identity and 86% similarity in amino acid sequences, and are expressed in highly overlapping cell types (http://travadb.org/), indicating high functional redundancy. A yeast two-hybrid assay was used to detect direct interactions between Msd1a (At5g57410), Msd1b (At2g18876), and Wdr8 (At5g07590). The results showed that Msd1a and Msd1b interacted with Wdr8 (Supplementary Fig. 1a). Strong auto-activation of Wdr8 as a bait precluded interaction assays between Wdr8 and its potential association partners. Next, we immunoprecipitated interacting proteins using a green fluorescent protein (GFP) antibody from transgenic *Arabidopsis* plants stably expressing GFP, Msd1b-GFP, or Wdr8-GFP (Fig. 1a and Supplementary Fig. 1b). From the Msd1b-GFP-expressing

plants, Msd1a, Msd1b, and Wdr8 were identified in the immunoprecipitates by liquid chromatography–tandem mass spectrometry (LC-MS/MS), in addition to Msd1b-GFP. Similarly, immune-precipitates from plants expressing Wdr8-GFP were also enriched in Msd1a, Msd1b, and Wdr8. These studies suggest that the in vivo protein complexes contain multiple copies of Wdr8 and Msd1, which may consist of either Msd1a, Msd1b, or both isoforms.

Previously, when GFP-fusions of putative *Arabidopsis* microtubule-associated proteins were transiently overexpressed in onion epidermal cells, Msd1a-GFP was localized along the lattice of cortical microtubules, whereas Wdr8-GFP was mostly found in the cytosol[23]. In this study, we expressed GFP-fusions of Msd1a, Msd1b, and Wdr8 under the regulation of their native promoters in transgenic *Arabidopsis* plants that expressed mCherry-β-tubulin 6 (TUB6). In pavement cells of cotyledons, GFP-fusions of Msd1a, Msd1b, and Wdr8 were all localized as relatively stable punctate particles on cortical microtubules (Fig. 1b). However, averaged projections of 151 frames (302 s) revealed that both Msd1 fusions also labeled the microtubule lattice, while a diffuse cytosolic pattern was revealed for Wdr8-GFP. These similar but distinct subcellular localizations of Msd1 and Wdr8 reflect our previous observations in the transient overexpression studies[23] and are consistent with the identification of *Xenopus laevis* SSX2IP (an ortholog of Msd1) as a microtubule-associated protein[5].

To gain further insights into the microtubule-related processes in which the plant Msd1–Wdr8 complex is involved, we investigated the localization of Msd1–Wdr8 in relation to the dynamics and nucleation events of cortical microtubules, as visualized with co-expression of mCherry-TUB6 and GFP-fusions of Msd1a, Msd1b, or Wdr8 (Fig. 1c, d and Supplementary Movies 1–4). In *Arabidopsis* interphase cells, microtubules are predominantly nucleated from γTuRCs that are recruited along the preexisting cortical microtubules[12,14]. Large fractions of Msd1-GFP and Wdr8-GFP puncta were observed at cortical nucleation sites (Msd1a, 47.4%, n = 962; Msd1b-GFP, 59.1%, n = 1198; and Wdr8-GFP, 60.7%, n = 1149). Most nucleation site-associated puncta were observed at branching nucleation sites (73–83%), followed by bundle-forming sites (parallel with mother microtubules, 15–23%) and a small number at free nucleation sites (<4%) (Fig. 1e). This distribution of localization is similar to the proportion of each type of nucleation site in these arrays[14,24] and highly similar to that of MOZART1a-GFP[25] (MZT1-GFP), an integral component of the γTuRC and essential for microtubule nucleation in *Arabidopsis*[25]. Punctate localization of Msd1a-GFP at nucleation sites was abolished in *wdr8* cells (69 events observed from 3 cells) and likewise for Wdr8 in *msd1a msd1b* double mutant cells (54 events observed from 3 cells) (Fig. 1c, d, Supplementary Figs. 2 and 3, and Supplementary Movies 5–8). Together, these observations show that Msd1 and Wdr8 localize to cortical microtubule nucleation sites, and that the function of both Msd1 and Wdr8 is required for such localization, supporting the idea that the observed punctate signals represent the heterodimeric complexes of Msd1 and Wdr8, and that stable localization to cortical nucleation sites requires complex formation.

To precisely determine the temporal localization of the Msd1–Wdr8 complex with respect to γTuRC, we co-expressed Msd1a-mCherry and MZT1-GFP in transgenic *Arabidopsis* plants. Confocal imaging of epidermal cells in the hypocotyl of dark-grown seedlings revealed that predominant portions of the Msd1a labels (99.0% in 110 particles in 3 cells) and the MZT1 labels (90.1% in 121 particles in 3 cells) colocalized with the other labels (Fig. 1f). When the exact timing of their cortex appearance was examined, the majority of Msd1a particles (76.8% in 341

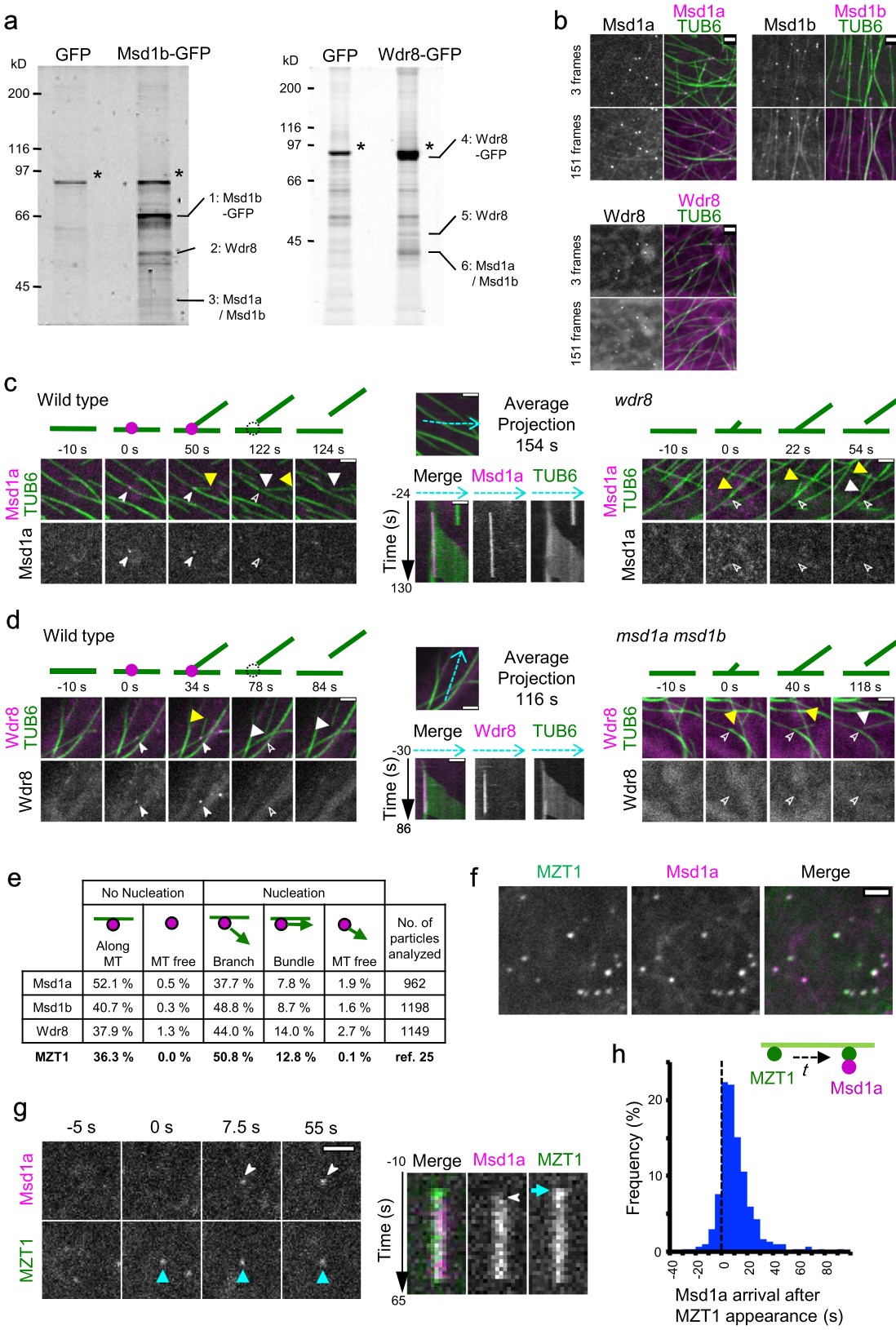

particles in 6 cells) appeared after a short delay to cortical locations where MZT1-labeled γTuRCs had already been recruited (mean time lag of 13.7 ± 2.8 s, Fig. 1g, h and Supplementary Movie 9). Both labels disappeared from the cell cortex approximately at the same time. These results suggest that shortly after γTuRC is recruited to cortical microtubules, the Msd1–Wdr8 complex becomes stably associated with it. The Msd1 ortholog in fission yeast is reported to interact with Alp4, a component of γTuRC[4].

**Release of daughter microtubules is delayed in *msd1* and *wdr8* mutant cells.** We found that the mean life time of MZT1-GFP

**Fig. 1 The Msd1–Wdr8 complex associates with the γ-tubulin ring complex at cortical nucleation sites along the microtubule lattice. a** Immuno pull-down experiments of Msd1- or Wdr8-interacting proteins from *Arabidopsis* seedlings. Seedlings stably expressing either GFP, Msd1b-GFP, or Wdr8-GFP were used to prepare cell extracts for immunoprecipitation using GFP-antibody beads. Precipitated proteins were separated by SDS-PAGE and detected by staining with Flamingo fluorescent dye. Asterisks indicate major nonspecific bands. This experiment was repeated twice with similar results. **b** Subcellular localizations of Msd1a-GFP, Msd1b-GFP, and Wdr8-GFP in cotyledon pavement cells when expressed under their native promoters. The TUB6 marker labels microtubules. Snap shots (3 frames) and integrated exposures of 151 frames (302 s total time) are shown. **c, d** Recruitment of Msd1a-GFP particles (**c**) and Wdr8-GFP particles (**d**) to the branch-forming nucleation sites on cortical microtubules in wild-type (left), *wdr8* (**c**, right), and *msd1a msd1b* (**d**, right) cells. Time-lapse confocal microscopy images are shown at the indicated times. For wild-type cells, kymographs of Msd1a-GFP (**c**) or Wdr8-GFP (**d**) and microtubules were generated along the dotted blue lines in the average projection images of 154 s and 116 s, respectively. Open and closed arrowheads indicate the absence and the presence, respectively, of Msd1a or Wdr8 particles. Likewise, the yellow and white triangles show the plus and minus ends of daughter microtubules, respectively. The events occurring at the indicated time points are schematically presented with microtubules (green lines) and Msd1–Wdr8 particles (magenta circles). **e** Localization of Msd1a-GFP, Msd1b-GFP, and Wdr8-GFP in relationship to microtubule nucleation was classified into five event groups. Percentages of events observed in five groups and the total number of observed events are shown. **f** Colocalization of Msd1a-mCherry and MZT1-GFP particles on the cell cortex region in *Arabidopsis* hypocotyl cells in the average projection images of 62.5 s. More than 110 particles from three cells were observed with similar results. **g** Recruitment of Msd1a-mCherry and MZT1-GFP particles. Left: time-lapse confocal microscopy images at the indicated times. Right: kymograph generated from the time-lapse microscopy images shown in the left. **h** Distribution of the arrival times (*t* in the diagram) of Msd1a-mCherry particles (magenta) after the appearance of MZT1-GFP particles (green) on the cell cortex ($n = 304$ events from 6 cells). The MZT1 particles arrive prior to the Msd1 particles. Bars, 2 μm.

particles at the cell cortex is considerably longer in the *msd1a msd1b* cells and the *wdr8* cells than in control cells (Fig. 2a and Supplementary Movie 10). The MZT1-GFP particles on the cortical microtubules that did not result in microtubule nucleation disappeared in a relatively short time (Fig. 2b); the residence time of these non-nucleating complexes did not significantly differ among wild-type cells ($14.0 \pm 20.7$ s, $n = 652$ particles), *msd1a msd1b* cells ($7.9 \pm 13.7$ s, $n = 554$ particles), and *wdr8* cells ($12.1 \pm 21.3$ s, $n = 340$ particles). By contrast, the microtubule-nucleating MZT1-GFP particles in the mutant cells remained on cortical microtubules longer than those in the wild-type cells (Fig. 2b). The proportions of the nucleation particles that persisted longer than 160 s were 3.4% in wild-type cells ($n = 378$), 39.4% in *msd1a msd1b* cells ($n = 226$), and 30.7% in *wdr8* cells ($n = 215$). To dissect nucleation and release events, we measured the time to nucleate microtubules after MZT1-GFP particles appeared on the lattice of cortical microtubules (Fig. 2c). The nucleation time was not significantly different among wild-type cells ($20.6 \pm 16.9$ s, $n = 235$ particles), *msd1a msd1b* cells ($25.5 \pm 19.7$ s, $n = 59$ particles), and *wdr8* cells ($17.9 \pm 14.4$ s, $n = 76$ particles). Thus, in the mutant cells, nucleated γTuRCs persisted much longer on mother microtubules than in wild-type cells.

Subsequent to nucleation of the daughter microtubule, the γTuRC in wild-type cells is destabilized and disappears from the mother microtubule by two distinct mechanisms[14]; release of the minus end of the daughter microtubule (Fig. 2d and Supplementary Movie 11) and complete shrinkage of the dynamic plus end of the daughter microtubule (Fig. 2e and Supplementary Movie 12). In wild-type cells, release events were observed to predominate (67.5% in 449 particles) over shrinkage events. By contrast, the majority of MZT1-GFP disappearance was caused by microtubule shrinkage in *msd1a msd1b* cells (74.9% in 223 particles) and *wdr8* cells (80.2% in 364 particles) (Fig. 2f). The observed shift of MZT1-GFP disappearance might be explained if microtubule release was less likely or more delayed in the mutant cells. To test this idea, we measured the time distribution of daughter microtubule release after nucleation. The release of daughter microtubules took a mean time of $63.8 \pm 32.3$ s ($n = 235$ events) in wild-type cells, whereas the mean release time increased significantly to $130.8 \pm 80.3$ s ($n = 59$ events) in *msd1a msd1b* cells and $157.7 \pm 171.3$ s ($n = 76$ events) in *wdr8* cells (Fig. 2g). Thus, extended persistence of γTuRC complexes observed in *msd1* and *wdr8* cells likely results from the delayed release of daughter microtubules from cortical nucleation sites,

implying that the Msd1–Wdr8 complex functions to promote daughter release.

**Recruitment of katanin to cortical nucleation sites requires Wdr8.** In interphase plant cells, katanin is known to sever microtubules at branching nucleation sites[14,15]. Therefore, a reasonable mechanism for Msd1–Wdr8 action on daughter microtubule release might be to recruit katanin or stimulate katanin activity. To investigate these possibilities, we first asked whether katanin is recruited to the subcellular sites where Msd1–Wdr8 particles have been localized. When dynamics of GFP-KTN1 (a GFP fusion of the p60 catalytic subunit, expressed from native upstream sequences) and Msd1a-mCherry were simultaneously followed in the wild-type cells, some GFP-KTN1 particles and some Msd1a-mCherry particles were transiently colocalized at the cell cortex (Supplementary Fig. 1 and Supplementary Movie 13). In these instances, the KTN1 particles arrived at the cortical locations where the Msd1a particles had already been tethered with a mean lag time of $19.1 \pm 21.5$ s ($n = 90$ events from 3 cells), indicating transient association of Msd1–Wdr8 with katanin.

Next, we examined whether katanin recruitment to branching nucleation sites is affected in the *wdr8* mutant cells. After nucleation of a daughter microtubule on the lattice of a mother microtubule in wild-type cells, fluorescence signals of GFP-KTN1 gradually increased at the base of the branched microtubules. Initial accumulation of GFP-KTN1 was weak and fluctuating, but over time the signal intensity incremented and peaked just before the observed detachment of the minus end (Fig. 3a, b, d and Supplementary Movie 14). The observation that GFP-KTN1 signal at nucleation sites was initially low and fluctuating before reaching a peak suggests that multiple copies of the katanin hexamer may be needed to completely sever microtubules from these sites. In many cases (67% of 30 events from 3 cells), GFP-KTN1 signals dissipated before detection of the minus-end shortening, or the two events were observed at the same time-lapse frames (30%). As the slowly depolymerizing minus end needs to shorten sufficiently to resolve optically the minus end from the mother microtubule, signal loss due to release of katanin complexes therefore likely occurred at or shortly after the minus-end detachment. Some GFP-KTN1 puncta (60% of 30 events from 3 cells) were dislodged from the nucleation sites prior to disappearance (Fig. 3b, arrowheads), indicating that association of katanin to the nucleation sites is destabilized shortly after the minus-end severing. By contrast, in the *wdr8* cells, GFP-KTN1

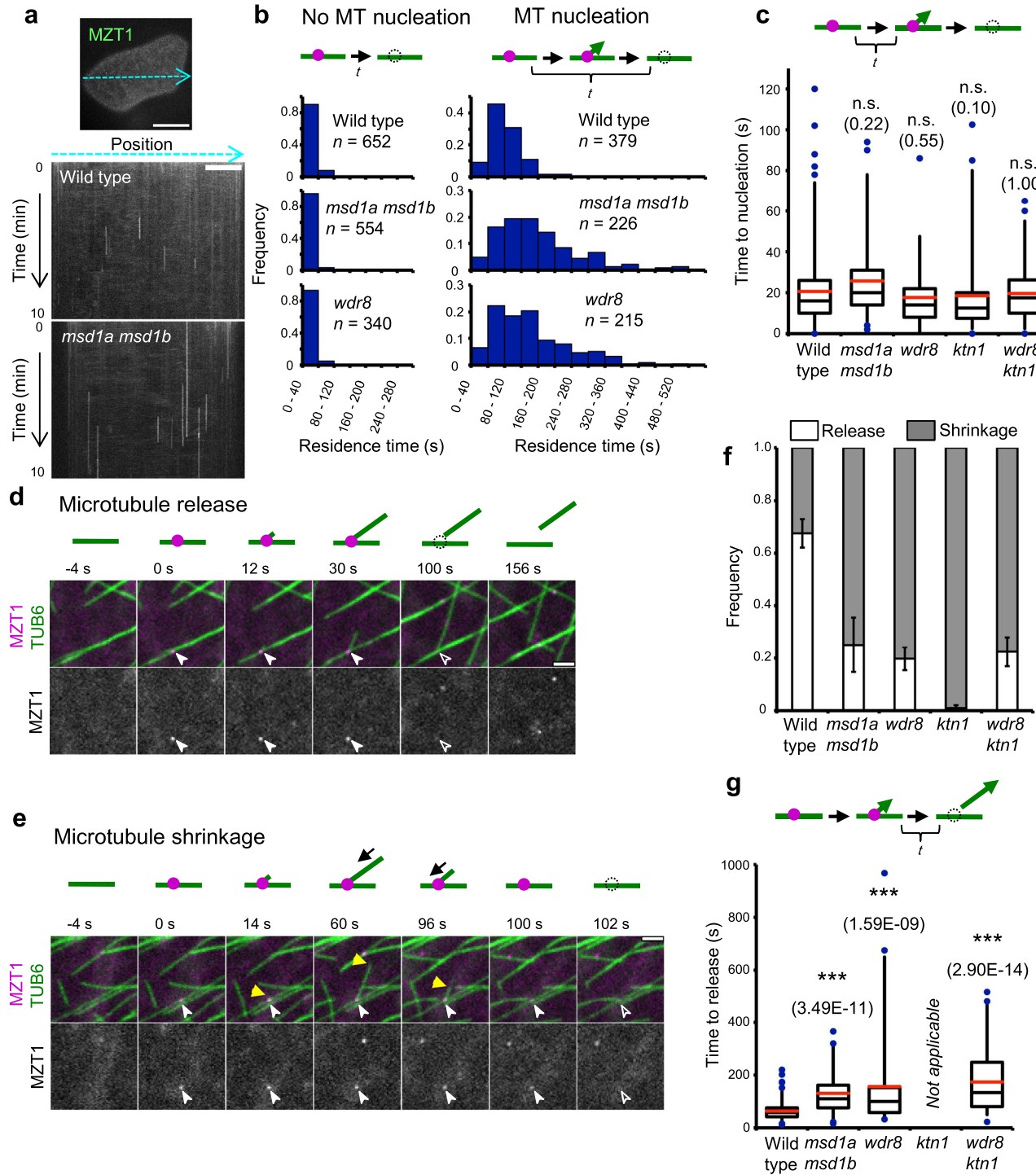

did not accumulate at the base of branched microtubules above the background level (Fig. 3c–e and Supplementary Movie 15). Thus, Wdr8 is required for katanin recruitment to the branching nucleation sites, explaining delayed daughter microtubule release.

As Msd1a-GFP, Msd1b-GFP, and Wdr8-GFP are all recruited to the bundle-forming nucleation sites in wild-type cells (Fig. 1e; also see Supplementary Fig. 5 and Supplementary Movies 16–18), we asked whether Msd1–Wdr8 is also necessary for katanin recruitment to these nucleation sites. Although bundle-forming nucleation is technically difficult to identify in cells expressing both mCherry-TUB6 and GFP-KTN1, we managed to find several such events with confidence (seven events in wild-type cells and three events in *wdr8* cells). In wild-type cells, GFP-KTN1 arrived at the

bundle-forming nucleation sites dozen seconds after nucleation in all seven cases (Fig. 3f and Supplementary Movie 19). In contrast, in *wdr8* cells, GFP-KTN1 particles were not detected at the minus end of daughter microtubules during the period of parallel microtubule nucleation and subsequent release (Fig. 3g and Supplementary Movie 20). Therefore, daughter microtubules nucleated by either branching or bundle-forming manner require Wdr8 to recruit katanin.

**The Msd1–Wdr8 complex stabilizes the association of daughter microtubule minus ends to their nucleation sites.** Despite the lack of detectable GFP-KTN1 recruitment at nucleation sites in the *wdr8*

**Fig. 2 Mutations of the Msd1–Wdr8 complex delay the time to daughter microtubule release after nucleation. a** Persistence of γTuRC, as labeled with MZT1-GFP, on the cell cortex region was followed over 10 min by time-lapse confocal microscopy in cotyledon pavement cells. Kymograph analysis was performed along images of the cell cortex as exemplified by the blue dotted line in this wild-type cell. Kymographs of a wild-type cell and an *msd1a msd1b* mutant cell are shown. Bar, 20 µm in the panel image and 10 µm in the kymograph. Three cells were analyzed with similar results. **b** Distribution of the residence times (*t* in the diagram) of the MZT1-GFP particles (magenta circles) on the mCherry-TUB6-labeled cortical microtubules (green lines) in wild type (seven cells), *msd1a msd1b* (four cells), and *wdr8* (five cells). The particles were classified into those that disappeared without or after microtubule nucleation. In the mutant cells, nucleating γTuRC particles tend to stay longer on the microtubule lattice. **c** Times to nucleate nascent microtubules (*t* in the diagram) from the lattice-bound γTuRC (magenta circles) in wild type (235 events from 6 cells), *msd1a msd1b* (58 events from 6 cells), and *wdr8* (76 events from 9 cells), *ktn1* (177 events from 4 cells), and *wdr8 ktn1* (66 events from 4 cells). Nucleating times of the mutants are not significantly different from that of wild type ($p > 0.09$, determined by the Steel–Dwass test). The bottom and top of each box represent the first and third quartiles, respectively, the horizontal line inside each box is the median (second quartile) and the red horizontal line represents the average. The whiskers range between 2nd percentile and 98th percentile. Data points outside this range are considered as outliers and are indicated as circles. n.s., not significant. Exact *p*-values are indicated in parentheses. **d**, **e** Time-lapse confocal microscopy images of the minus-end release event (**d**) and the complete shrinkage event from the plus end (**e**) of daughter microtubule at the indicated times in wild-type cells. γTuRC (magenta circles in the diagrams) and microtubules (green lines) are labeled by MZT1-GFP and mCherry-TUB6, respectively. Open and closed arrowheads and yellow triangles respectively indicate the absence and the presence of MZT1 particles and the plus ends of daughter microtubules. Bars, 2 µm. **f** Proportions of the MZT1 particle disappearance caused by the minus-end release (green) and the plus-end shrinkage (magenta) in wild-type (449 events from 7 cells), *msd1a msd1b* (223 events from 6 cells), and *wdr8* (367 events from 7 cells), *ktn1* (301 events from 4 cells), and *wdr8 ktn1* (294 events from 4 cells). The error bars indicate SD. **g** Times (*t* in the diagram) for daughter microtubules to be released from mother microtubules after nucleation. The event numbers for each genotype are the same with **c**. In the diagram, a daughter microtubule grows at the plus end (arrowhead) after nucleation from γTuRC (magenta). ***$p < 0.0001$, determined by the Steel–Dwass test. Exact *p*-values are indicated in parentheses. The analysis is not applicable for the *ktn1* cells where the release events are very rare. The elements of the box plot are defined as in **c**.

mutant cells, the minus ends of daughter microtubules were eventually detached, indicating that katanin is not absolutely required for the minus-end detachment in the *wdr8* cells. This observation was surprising, as our previous investigations[14,15] had shown that katanin was required for the release of daughter microtubules from nucleation complexes by severing. To further characterize the katanin-independent microtubule detachment events observed in the *wdr8* cells, we directly compared microtubule release events in p60 katanin mutant (*ktn1*) cells and *wdr8 ktn1* double mutant cells. Consistent with previous observations, in the *ktn1* cells, daughter microtubules generally remained attached to the lattice of mother microtubules at the branched nucleation sites during the observation period of >15 min, unless the mother microtubule depolymerized (Fig. 4a and Supplementary Movie 21). In rare cases, daughter microtubules were found to be released eventually (three detachments events observed after 125, 163, and 750 s of microtubule nucleation in a total of 301 branching nucleation events observed), concomitant with a disappearance of the MZT1-GFP nucleation complex marker (Supplementary Fig. 6 and Supplementary Movie 22). In the *wdr8 ktn1* cells, on the other hand, daughter microtubules were now readily released (Fig. 4b and Supplementary Movie 23), along with loss of MZT1-GFP (22.4% of observed branching nucleation events over the observation interval). This rate of daughter detachment and MZT1-GFP loss in the absence of daughter microtubule shrinking was comparable to that observed in the *wdr8* single mutant cells (19.8%) (Fig. 2f). The MZT1-GFP particles were released in conjunction with daughter detachment after 174.2 s (±123.7 s in 63 events) of microtubule nucleation in the *wdr8 ktn1* cells, which was also comparable with the release time in the *wdr8* cells (Fig. 2g). These results indicate that Wdr8 is required to stabilize the nucleated branching sites on cortical microtubules. Without Wdr8, the branched nucleation sites readily disintegrate and the minus ends of daughter microtubules are released even in the absence of the severing activity of katanin. Thus, it appears that Wdr8 has two functions; it is needed to stabilize the nucleated γTuRC on the side of the mother microtubule and it is required to recruit katanin to detach the daughter microtubule after it is nucleated.

We further examined whether other nucleation characteristics of cortical microtubules are affected in the mutant cells. Nucleation frequencies per unit cortical area per unit time (sum of branching,

bundle-forming, and free nucleation types) were not significantly different ($P > 0.6$, Steel–Dwass test) between genotypes ($0.43 \pm 0.09$ per 100 µm² per min, $n = 541$ from 7 wild-type cells, $0.47 \pm 0.22$ per 100 µm² per min, $n = 355$ from 6 *mds1a msd1b* cells, and $0.42 \pm 0.24$ per 100 µm² per min, $n = 309$ from 5 *wdr8* cells). In the branch-forming nucleation, the angles formed between mother microtubules and daughter microtubules were not significantly different between wild-type cells ($42.2 \pm 9.7$, $n = 126$ events from 4 cells), *msd1a msd1b* cells ($38.5 \pm 13.0$, $n = 121$ from 5 cells), and *wdr8* cells ($39.0 \pm 14.9$, $n = 97$ from 5 cells) (Supplementary Fig. 7a). In the wild-type cells, the branch-forming nucleation predominates over the bundle-forming parallel nucleation[13,14]. In the *msd1a msd1b* cells ($n = 355$ events from 6 cells) and *wdr8* cells ($n = 309$ events from 5 cells), the relative proportion of the bundle nucleation type moderately but significantly increased over the branching nucleation type, compared to the wild-type cells ($n = 541$ events from 7 cells) ($P < 0.001$, Fisher's exact test) (Supplementary Fig. 7b, c). The Msd1–Wdr8 complex may have larger roles in stabilizing the branch-forming orientation, compared to parallel orientation, of γTuRC on the cortical microtubules.

**Katanin recruitment during cell division does not require the Msd1–Wdr8 complex.** Plant microtubules in mitotic arrays, spindles, and phragmoplasts, are also nucleated along preexisting microtubules in a γTuRC- and augmin-dependent manner[26,27], and katanin is required for the organization of mitotic microtubule arrays and for proper progression of mitosis and cytokinesis[21,22,28]. Subcellular localization of Msd1a-mCherry and GFP-KTN1 during mitosis revealed partially overlapping but distinct distributions. In particular, Msd1a-mCherry localized broadly on prophase and metaphase microtubules, whereas katanin mostly accumulated after anaphase and distributed more toward the spindle poles where the minus ends are enriched (Fig. 5a and Supplementary Movie 24). Katanin localization on mitotic microtubules was indistinguishable between wild-type cells and the *wdr8* mutant cells (Fig. 5b and Supplementary Movies 25 and 26). Therefore, Msd1–Wdr8 does not appear to play a major role for katanin recruitment on mitotic microtubules. A phragmoplast-localized plant-specific protein CORD4 contributes to tethering of katanin to the distal end of phragmoplast microtubules[28].

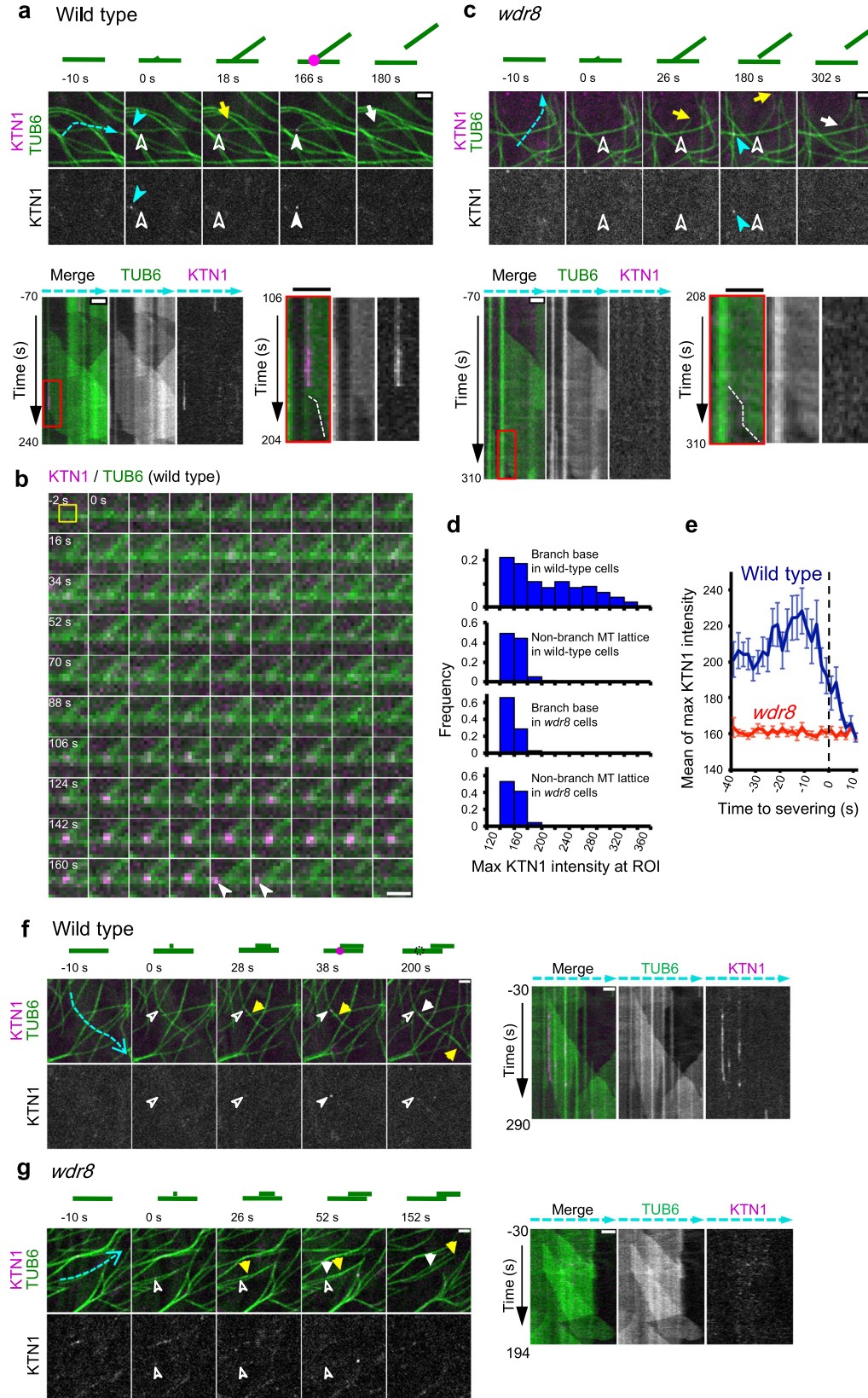

**Katanin recruitment to microtubule crossover sites does not require the Msd1–Wdr8 complex**. GFP-KTN1 was recruited also to microtubule crossover sites at the cell cortex (Fig. 6a–d and Supplementary Movies 27 and 28). Katanin consistently accumulated at crossover sites prior to severing, consistent with

previous observations[17,18]. After severing, the new plus end first underwent rapid depolymerization. Slow depolymerization of the minus end was observed several seconds later (5.4 ± 5.6 s in 25 events), whereas GFP-KTN1 signal sometimes remained for several seconds at the severed crossover site. Recruitment of GFP-

**Fig. 3 Wdr8 is required to recruit katanin to the cortical nucleation sites. a, c** Time-lapse confocal microscopy images of katanin recruitment to the branching nucleation sites at the release of daughter microtubules in wild-type (**a**) and *wdr8* (**c**) cells. Katanin (magenta circles in the diagrams) and microtubules (green lines) are labeled by GFP-KTN1 and mCherry-TUB6, respectively. Open and closed white arrowheads respectively indicate the absence and the presence of KTN1 particles, whereas yellow and white arrows show the plus and minus ends of daughter microtubules. Closed blue arrowheads point to KTN1 particles at the microtubule crossover sites. Kymographs track the dotted blue lines. Microtubule nucleation events set the zero time points. Rectangle regions boxed by red lines on the left kymographs are enlarged on the right where dotted white lines mark the minus ends of released daughter microtubules. Bars, 2 μm. The images in **a** are representative of 90 similar events, whereas those in **c** represent 29 similar events. **b** Time-lapse images of katanin (magenta) and microtubules (green) at 2 s intervals in wild-type cells. Microtubule nucleation occurs at 0 s, whereas detachment of the daughter microtubule minus end is recognizable at 172 s. White arrowheads indicate the katanin particles dislodged from the branch base. A yellow square at −2 s shows ROI (5 pixel × 5 pixel). Bar, 1 μm. The images are representative of 30 similar events. **d** Distribution of GFP-KTN1 signal intensities at the branching nucleation sites (384 ROIs from 6 wild-type cells and 464 ROIs from 4 *wdr8* cells) and the non-nucleating regions of the mother microtubule lattice (272 ROIs from 6 wild-type cells and 320 ROIs from 4 *wdr8* cells). MT, microtubule. **e** Katanin accumulates at the branching nucleation sites prior to severing in wild-type cells (green line; 24 events from 6 cells) but not in *wdr8* cells (magenta line; 29 events from 4 cells). Detachment of the minus ends of daughter microtubules from the mother microtubules were visually detected from the confocal images and was set to the time zero. The error bars indicate SD. **f, g** Time-lapse confocal microscopy images of katanin recruitment to the bundle-forming nucleation sites at the release of daughter microtubules in wild-type (**f**) and *wdr8* (**g**) cells. Similar events were observed in nine wild-type cells (**f**) and three *wdr8* cells (**g**). Kymographs on the right track the dotted blue lines. Symbols are the same as in **a** and **c**. Bars, 2 μm.

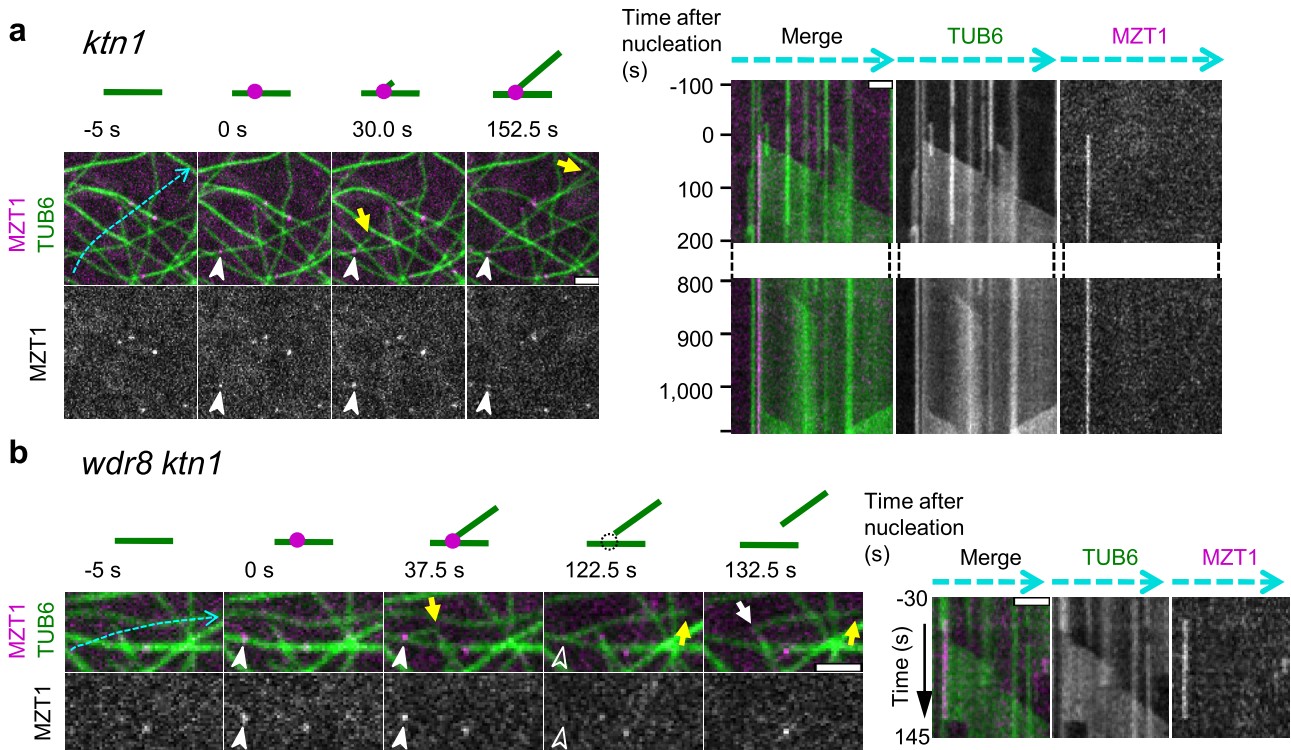

**Fig. 4 Wdr8 stabilizes the branching nucleation sites.** Time-lapse confocal microscopy images of the minus-end stability of daughter microtubules after branch-forming nucleation. γTuRC (magenta circles in the diagrams) and microtubules (green lines) are labeled by MZT1-GFP and mCherry-TUB6, respectively. Open and closed arrowheads respectively indicate the absence and the presence of MZT1 particles, whereas yellow and white arrows show the plus and minus ends of daughter microtubules. Kymographs track the dotted blue lines. Appearance of MZT1 particles set the zero time point. **a** In the *ktn1* cells, the minus ends of daughter microtubules are stably attached to the branching nucleation sites, as seen in the kymograph at 1100 s after nucleation. A total of 298 similar events were observed. **b** In the *wdr8 ktn1* cells, daughter microtubules are readily released even in the absence of katanin. A total of 63 similar events were observed. Bars, 2 μm.

KTN1 and severing of crossed over microtubules did not differ between wild-type cells and the *wdr8* cells; 94.4% of the severed crossover sites in wild-type cells (*n* = 54, 22 cells) and 95.4% of the sites in the *wdr8* cells (*n* = 65, 20 cells) showed evidence of prior accumulation of katanin. In support of this, microtubule crossover sites in wild-type cells generally did not show recruitment of Msd1a-GFP (96.2% of 105 particles from 10 cells) and Wdr8-GFP (97.8% of 91 particles from 10 cells) before the severing events (Fig. 6e, f and Supplementary Movies 29 and 30). Thus, Msd1 and Wdr8 do not play essential roles in either katanin recruitment or activity at microtubule crossovers.

**Mutations in *Msd1* or *Wdr8* rescue interphase cell expansion phenotypes of *katanin* mutant plants.** We examined the requirements of Msd1–Wdr8 and katanin on growth and cell shape in *Arabidopsis* plants. Growth of whole seedlings (Fig. 7a) and mature plants (Fig. 7b) was almost indistinguishable in wild type, *msd1a msd1b*, and *wdr8*, but was severely inhibited in *ktn1*. Interestingly, the severe growth defects observed in *ktn1* were partially rescued by *msd1a msd1b* or *wdr8*. The growth rate of seedling primary roots was also highly impaired in *ktn1*, compared to wild type, *msd1a msd1b*, and *wdr8*, and was also significantly (although not completely) rescued in *msd1a msd1b*

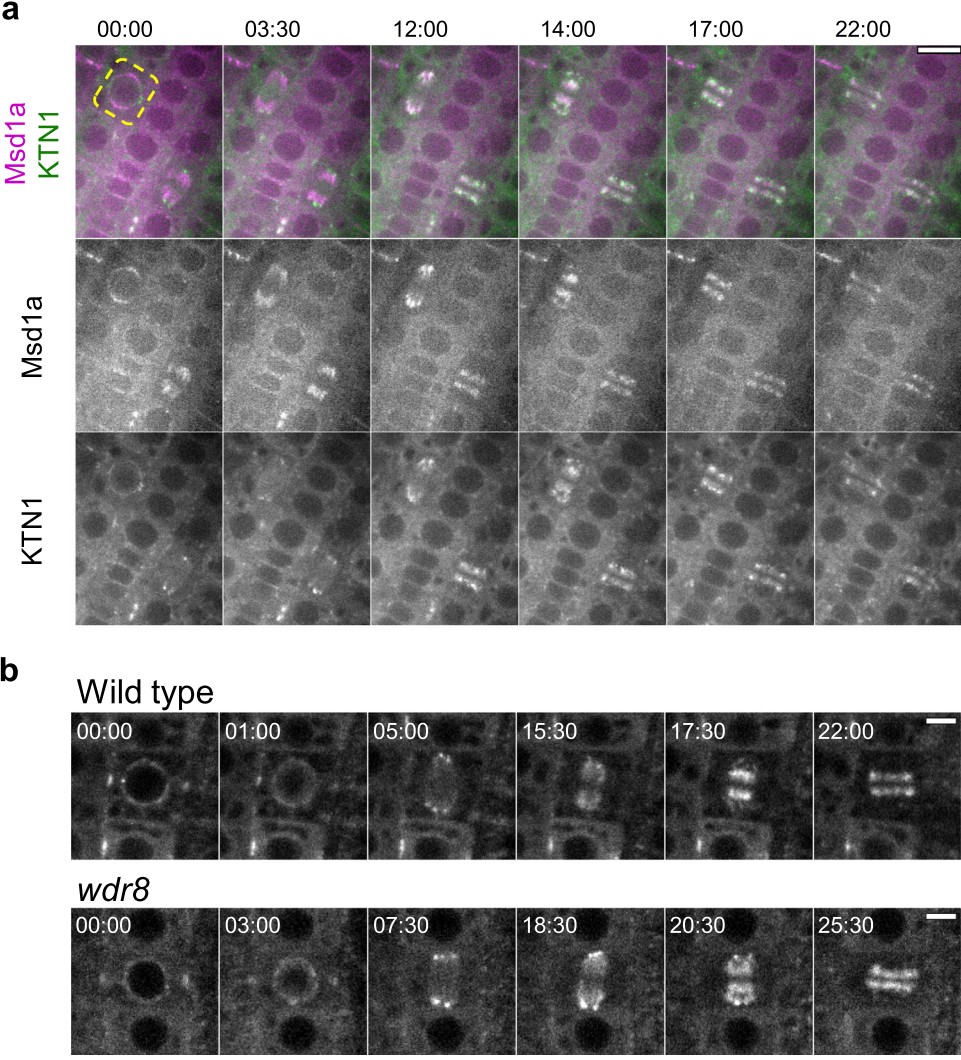

**Fig. 5 Wdr8 is not required for katanin localization during cell division. a** Colocalization of katanin and Msd1 during cell division. Time-lapse confocal microscopy images of GFP-KTN1 (green) and Msd1a-mCherry (magenta) in the dividing epidermal cells of the primary roots of 3-day-old wild-type seedlings. Beginning of the nuclear envelop breakdown in the root cell marked by the dotted yellow line sets the time zero points (minute : second). Katanin is substantially colocalized with Msd1 but accumulates more toward the minus ends of microtubules in spindles (12 : 00) and phragmoplasts (17 : 00). Four cytokinesis events from three plants were observed with similar results. Bar, 10 μm. **b** Subcellular localization of katanin during cell division. Time-lapse confocal microscopy images of GFP-KTN1 in the dividing epidermal cells of the primary roots of 3-day-old wild-type and *wdr8* seedlings. Beginning of the nuclear envelop breakdown sets the time zero points (minute : second). Each images were averaged images of three *z*-slices taken with 0.7 μm step. A total of ten and eight similar events were observed in wild-type cells (from eight plants) and *wdr8* cells (from eight plants), respectively. Bars, 5 μm.

*ktn1* and *wdr8 ktn1* (Fig. 7c). As root growth results from mitotic cell divisions in the meristem and subsequent rapid cell elongation during interphase, we investigated both phenotypes individually. In the *ktn* mutants, cytokinesis was severely impaired in the meristem as manifested by skewed cell division planes[29]. In the pericycle cell files of the primary roots, the katanin defect did not affect the mean angle between the transverse cell wall and the cell file's axis but significantly increased the variance of the orientation angles (Supplementary Fig. 8). The *msd1* and *wdr8* mutations alone did not significantly affect cell division orientation and did not complement the cytokinetic defects of *ktn1*. In interphase plant cells, cortical microtubule arrays are organized transverse to the longitudinal axis of axial organs and are required for anisotropic cell elongation, at least in part by promoting ordered cellulose deposition[30,31]. Confocal cross-sections of primary roots showed that cortical cells at the elongation zone and at the differentiation zone adopted long cylindrical shapes in wild type, *msd1a msd1b*, and *wdr8*, but were radially swollen in *ktn1*

(Fig. 7d). The isotropic cell expansion phenotype in *ktn1* was substantially rescued by genetic combination with *msd1a msd1b* or *wdr8* (Fig. 7e). Taken together, these observations suggest that katanin-mediated severing antagonizes Msd1–Wdr8-mediated stabilization of interphase nucleation structures to allow for appropriate cortical microtubule array organization and postmitotic anisotropic expansion of plant cells.

## Discussion
Our results show that in acentrosomal plant cells, the evolutionary conserved Msd1–Wdr8 complex anchors γTuRC to branching nucleation sites on interphase microtubules (Fig. 8). The absence of this complex does not affect the residence time of non-nucleating γTuRC complexes nor the time for γTuRC complexes to nucleate following their arrival on the mother microtubule lattice; however, it decreases the percentages of branch-forming nucleation with respect to bundle-forming nucleation, indicating that the Msd1–Wdr8 complex specifically

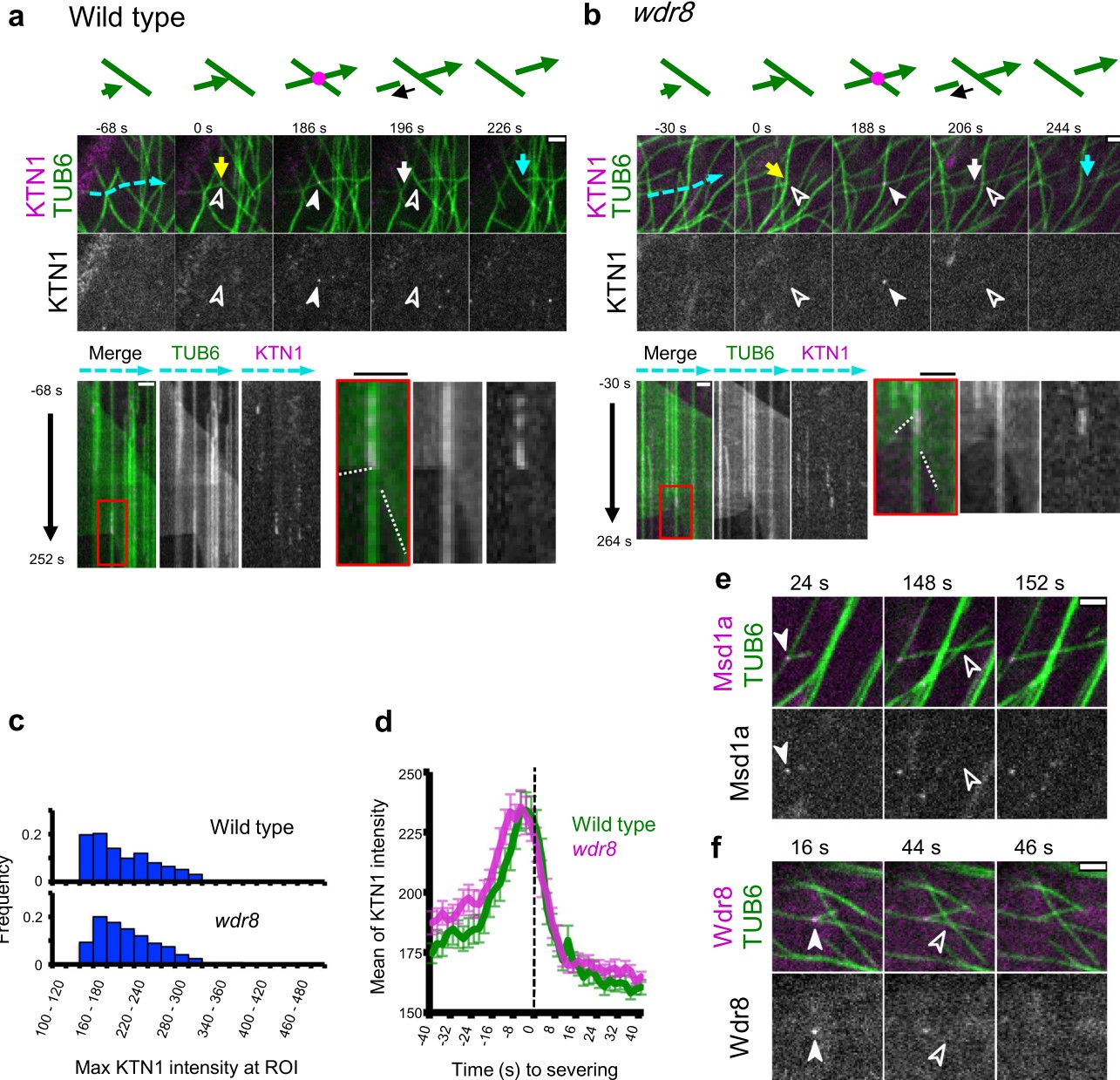

**Fig. 6 Katanin recruitment to the microtubule crossover site does not require Msd1–Wdr8. a**, **b** Time-lapse confocal microscopy images of katanin recruitment to the crossover sites in wild-type (**a**) and *wdr8* (**b**) cells. Katanin (magenta) and microtubules (green) are labeled by GFP-KTN1 and mCherry-TUB6, respectively. Open and closed arrowheads respectively indicate the absence and the presence of KTN1 particles. Yellow and white arrows show the growing and shrinking plus ends, whereas blue arrows indicate the shrinking minus ends of cortical microtubules. Kymographs track the dotted blue lines. Microtubule crossover events set the zero time points. Rectangle regions boxed by red lines are enlarged on the right where dotted white lines mark the plus (+) and minus (−) ends of severed microtubules. A total of 54 and 65 similar events were observed in wild-type cells and *wdr8* cells, respectively. Bars, 2 μm. **c** Distribution of GFP-KTN1 signal intensities at the crossover in wild type (720 ROIs from 14 cells) and *wdr8* (960 ROIs from 11 cells). **d** Katanin accumulates at the crossover sites prior to severing in wild type (green; 42 events from 14 cells) and in *wdr8* (magenta; 51 events from 11 cells). Detachment of the severed microtubules were visually detected from the confocal images and was set to the time zero. The error bars indicate SD. **e**, **f** Time-lapse confocal microscopy images of Msd1a-GFP (**e**) and Wdr8-GFP (**f**) (magenta), and mCherry-TUB6-labeled microtubules (green). Open and closed arrowheads respectively indicate the absence and the presence of Msd1/Wdr8 particles. Bars, 2 μm. **e** A daughter microtubule was formed at a branching nucleation site (labeled by Msd1a-GFP) and crossed bundled microtubules and another microtubule at 148 s, which resulted in severing without prior recruitment of Msd1a (152 s). **f** A daughter microtubule was formed at a branching nucleation site (labeled by Wdr8-GFP) and crossed a microtubule at 16 s, which resulted in severing in the absence of Wdr8 at 44 s. Nucleation of the daughter microtubule is set to time zero.

stabilizes the γTuRC on the microtubule lattice once it has nucleated a branched daughter microtubule. Diffuse association of the Msd1 subunit with the microtubule lattice and the slight but significant time lag of the appearance of punctate Msd1 and Wdr8 at γTuRC-labeled cortical nucleation sites suggest that the

complex either specifically assembles and accumulates at these locations or that the complexes are assembled on the lattice in general and then transported to the lattice-bound γTuRC. In fission yeast, a minus-end directed kinesin Pkl1 is required to transport the Msd1–Wdr8 complex along the spindle

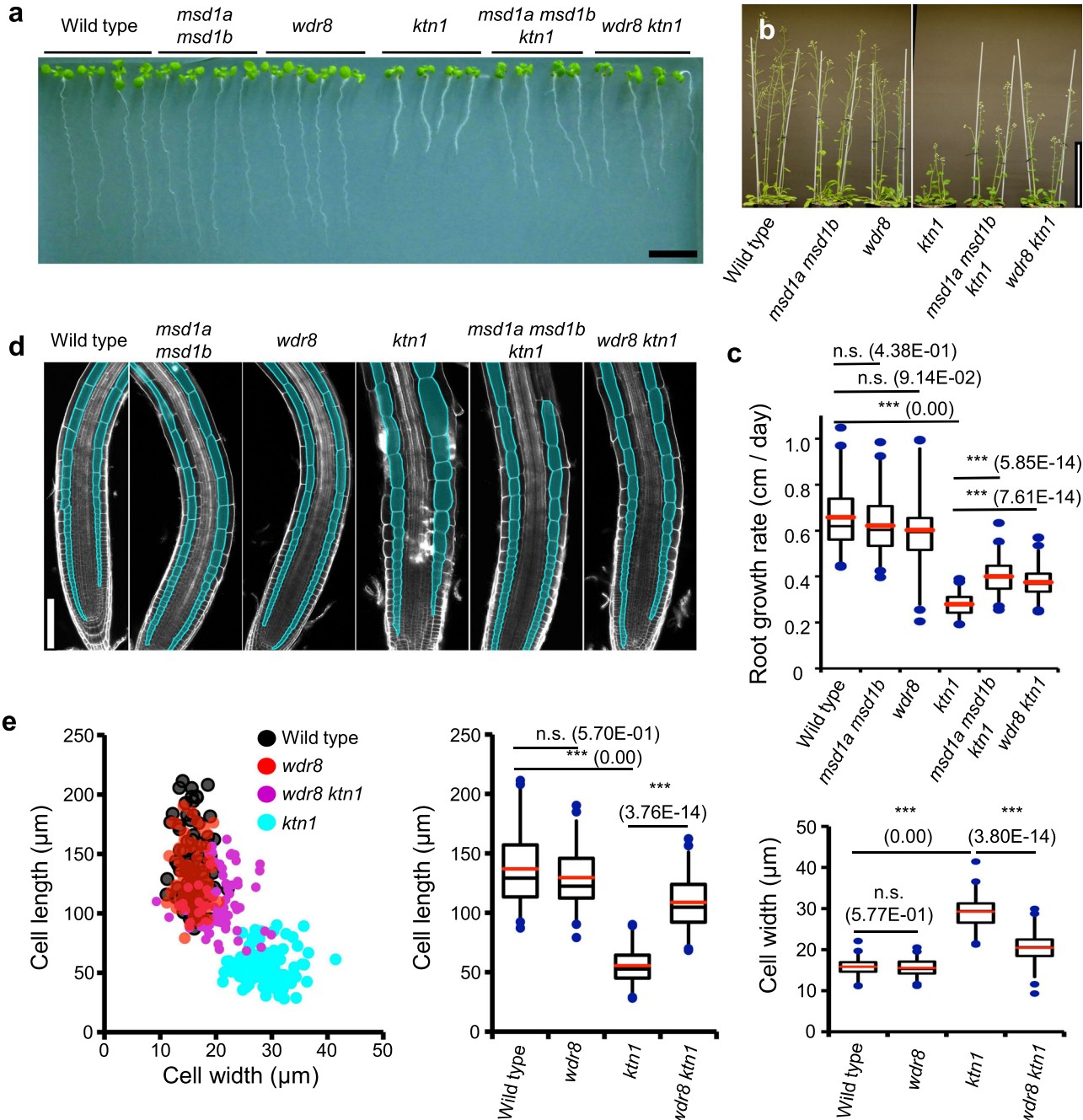

**Fig. 7 Mutations in the Msd1–Wdr8 complex complement the cell shape defects of katanin mutants. a** Eight-day-old Arabidopsis seedlings grown on a vertically placed agar medium. Bar, 10 mm. **b** Growth and statute of flowering *Arabidopsis* plants. Bar, 10 cm. **c** Growth rate (cm per day) measured by daily increment in the length of primary roots from 5 to 10 days after germination in wild type (20 seedlings), *msd1a msd1b* (20 seedlings), *wdr8* (20 seedlings), *ktn1* (19 seedlings), *msd1a msd1b ktn1* (18 seedlings), and *wdr8 ktn1* (18 seedlings). The bottom and top of each box represent the first and third quartiles, respectively; the horizontal line inside each box is the median (second quartile) and the red horizontal line represents the average. The whiskers range between 2nd percentile and 98th percentile. n.s., not significant and ***p < 0.0001, determined by the Steel–Dwass test. Exact *p*-values are indicated in parentheses. **d** Root tip morphology of indicated genotypes. Cortex cell files are highlighted with the turquoise paint. Bar, 100 μm. **e** Cortex cell shape of 4-day-old primary roots at the differentiation zone where cell expansion has ceased in wild type (98 cells from 4 seedlings), *wdr8* (47 cells from 3 seedlings), *ktn1* (94 cells from 2 seedlings), and *wdr8 ktn1* (76 cells from 2 seedlings). Left, cell length and cell width of individual cells. Middle, cell length. Right, cell width. n.s., not significant and ***p < 0.0001, determined by the Steel–Dwass test. Exact *p*-values are indicated in parentheses. The elements of the box plot are defined as in **c**.

microtubule to the spindle-pole body[32]. *Arabidopsis* microtubule motors might function in a similar manner.

How katanin is recruited to specific cellular addresses to sever microtubules is poorly understood. In a previous study, purified katanin complexes of *Caenorhabditis elegans* preferentially sever at intersections between two microtubules in vitro[33], suggesting that specific recruitment factors may not always be necessary. Here we have shown that Msd1–Wdr8 not only functions to stabilize branching nucleation structures in cortical arrays of *Arabidopsis* cells, but that it also functions as an essential factor to

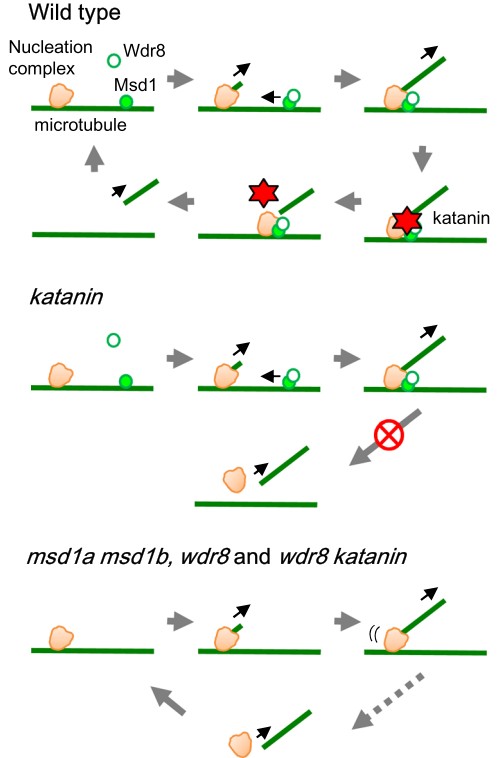

**Fig. 8 Model for stabilization of branching nucleation sites and recruitment of katanin by the Msd1–Wdr8 complex.** In wild-type cells, lattice-bound Msd1 (filled green circle) recruits cytoplasmic Wdr8 (open green circle) to form a heteromeric complex, which is translocated to and associated with γTuRC (orange) on a preexisting microtubule (green line). After nucleation of a microtubule, katanin is recruited by Msd1–Wdr8 and severs the minus end of the daughter microtubule. Direct physical interactions between katanin and Msd1–Wdr8 are yet to be demonstrated. In *msd1* and *wdr8* cells, nucleated γTuRC is not stably tethered to the mother microtubule and is eventually dissociated in the absence of the katanin activity. The *wdr8* mutation (and probably the *msd1* mutation as well) is epistatic to the *katanin* mutation. The figure was prepared originally by the authors.

recruit katanin to these sites for their disassembly. Such counteracting dual functions on the nucleation site stability likely underlie apparently normal growth of *msd1* and *wdr8* mutant plants under standard growth conditions. An analogous stabilization-destabilization mechanism of microtubule organization has been reported for mammalian microtubule minus-end-regulating proteins CAMSAP1 and CAMSAP2, which form stretches of stabilized microtubule lattice at the minus ends, and recruit katanin to antagonize the formation of long CAMSAP-decorated stretches[34].

The plant Msd1–Wdr8 complex would endow plant cells with the ability to place release of daughter microtubules under the strict regulation of the katanin activity. Spindle length in *Xenopus* species is controlled by inhibitory phosphorylation on the katanin p60 subunit[35]. Strict dependency of microtubule severing on katanin activity may make remodeling of microtubule array organization adaptable to changing cellular status and environmental stimuli. Although it is not well understood what developmental and environmental cues control the katanin activities in plant cells, the *msd1* and *wdr8* mutants, in which daughter microtubules are strictly released by the katanin-independent manner, may be useful for exploring environmentally stimulated

microtubule array remodeling processes in which katanin activation plays a critical role.

## Methods

**Plant materials and growth conditions.** *A. thaliana* ecotype Columbia-0 was used and the original plants and their transgenic derivatives expressing marker proteins were all referred to as the wild type in this study. Seedlings were grown on 1.5% agar medium containing a half-strength Murashige and Skoog salt mixture (pH 5.7) and 1% sucrose at 22 °C under a long-day photoperiod (16 h light/8 h dark) condition or in the dark. T-DNA tagged lines of *Msd1a* (*msd1a-1*, SALK_148799; *msd1a-2*, GABI_557F06), *Msd1b* (*msd1b-1*, SALK_132426; *msd1b-2*, GABI_596B10), and *Wdr8* (*wdr8-1*, SALK_093768) were obtained from the Arabidopsis Biological Resource Center (https://abrc.osu.edu/). To confirm the T-DNA insertion sites (Supplementary Fig. 2), genomic regions flanking the insertion sites were amplified by PCR using the primer sets listed in Supplementary Table S1 and were sequenced. The *ktn1-2* allele was reported previously[36].

**Transgenic plant lines expressing marker proteins.** The Msd1a-GFP, Msd1b-GFP, Wdr8-GFP, and Msd1a-mCherry constructs were generated by amplifying the corresponding genomic fragments of *Msd1a* (At5g57410; 7157 bp), *Msd1b* (At2g18876; 6952 bp), and *Wdr8* (At5g07590; 5797 bp) with the primers listed in Supplementary Table 1. GFP and mCherry were fused to the gene of interest separated by a GGGGSGGGGSGGGGS-linker[37], just before the stop codon. These fluorescent protein-fusion constructs were transferred to pBIN40[38] (with a hygromycin-resistance gene) or pBIN30[39] (with a bialaphos-resistance gene), which had been modified from pBIN19. An Agrobacterium-mediated transformation method[40] was used to generate transgenic plants. The transgenic plant lines expressing the fusion proteins were crossed with a microtubule-marker line expressing mCherry-TUB6[41] for co-visualization with microtubules. An Msd1a-mCherry transgenic plant was crossed with an MZT1a-GFP line[42] for colocalization analysis. A list of primers used for molecular cloning is provided in Supplementary Table 1.

To facilitate generation of dual marker lines in various mutant backgrounds, we constructed a microtubule nucleation marker binary vector, which harbors both an mCherry-TUB6 cassette for microtubule labeling and an MZT1a-GFP cassette for γTuRC visualization. After a StuI cloning site was introduced adjacent to the multi-cloning site of pBIN40, the UBI10 promoter-driven mCherry-TUB6-expressing cassette[41] was cloned into the StuI site and the MZT1a-GFP-expressing cassette[42] was then cloned into the AscI site of the resulting pBIN4100 plasmid. This binary vector was used to transform wild-type *Arabidopsis* plants to establish a microtubule nucleation dual marker line, which was then crossed to mutant plants. Marker lines with homozygous mutations for *msd1a-1 msd1b-1*, *wdr8-1* or *ktn1-2* were identified in subsequent generations. To co-visualize katanin and microtubules, the GFP-KTN1-expressing plant in the *ktn1-2* background[43] and the mCherry-TUB6 line were independently crossed with *wdr8-1*, and the progeny lines harboring both constructs in the *ktn1-2* and *ktn1-2 wdr8-1* homozygous backgrounds were identified in subsequent generations.

**Immunopurification and mass spectrometry.** Methods to purify GFP-tagged proteins and associated proteins from *Arabidopsis* plant extracts were described previously[36,42]. In brief, crude protein extracts in the extraction buffer (50 mM HEPES, 150 mM NaCl, 1% NP-40, 2 mM phenylmethylsulfonyl fluoride, Complete Protease Inhibitor EDTA-free cocktail (Roche), pH 7.5) were prepared from 10- or 12-day-old light-grown transgenic plants expressing a target GFP-fused protein. Immunoprecipitation was carried out using the μMACS GFP-tag protein purification kit (#130-091-125; Miltenyi Biotec, Bergisch Gladbanch, Germany), basically by following the manufacturer's protocol, except that the above extraction buffer was used in the first washing step. This kit includes μMACS Microbeads that are conjugated to an anti-GFP monoclonal antibody. Purified proteins were separated by SDS-polyacrylamide gel electrophoresis, stained with SYPRO Rubby (Thermo Fisher), and pooled into several fractions corresponding to their molecular weight. In-gel protein digestion to peptides and subsequent LC-MS/MS analysis was performed as reported[36].

**Time-lapse imaging.** Dynamics of Msd1, Wdr8, MZT1, and KTN1 with respect to locations of cortical microtubules in interphase cells were analyzed in adaxial pavement cells of 13- to 15-day-old cotyledons. Excised cotyledons were mounted on slide glasses with shallow holes (TOSHINRIKO CO., LTD) and cover glasses were immobilized on slide glasses with commercial nail polish to minimize drifting of specimens during observation. Images in Figs. 1–3 were obtained using a confocal microscope system previously reported[44]. mCherry and GFP were sequentially excited with 100 and 200 ms exposure, respectively, and images from three consecutive excitations were averaged. Images were acquired every 2 s for 5–15 min. For Fig. 4, an Olympus IX-83 inverted microscope equipped with a spinning disk confocal unit Yokogawa CSU-W1, a dichroic mirror DM405/488/561, an Andor EM-CCD camera iXron3 888, and a ×100, 1.40 numerical aperture (NA) UPlanSApo oil-immersion objective (Olympus) were used. mCherry and GFP were sequentially excited by 561 and 488 nm OPSL lasers (Coherent) for 400 and 800

ms, and images were collected every 2.5 s for 10 min with 617/73 and 520/35 emission filters, respectively.

Colocalization of Msd1a-mCherry and MZT1a-GFP (Fig. 1f) was analyzed in hypocotyl epidermal cells of 3-day-old etiolated seedlings with a confocal microscope system reported previously[36]. Specimens were prepared by the agar pads method as previously reported[45]. Both GFP and mCherry were excited with 800 ms exposures, and images were taken with 2.5 s intervals for 15 min. Two experiments with reverse orders of excitations were performed and the data were pooled.

Time-lapse imaging of mitotic localization of Msd1 and KTN1 in root tip cells (Fig. 5a) was done in glass-based dishes (IWAKI, #3910-035). Four-day-old seedlings were mounted on dishes with nutrition-containing agar block. A previously reported microscope system[44] was used with a ×60, 1.49 NA CFI Apochromat TIRF oil-immersion objective. mCherry and GFP were excited in this order with 200 and 150 ms exposure. Images were taken with 30 s intervals. Mitotic KTN1 localization in wild-type and *wdr8* mutant backgrounds (Fig. 5b) used the same microscope system as Fig. 4, with a ×60, 1.30 NA UPlanSApo silicon-immersion objective. Images were taken as 0.5 μm step z-stacks and with 30 s intervals.

**Image processing and analysis.** Image processing and analyses were performed with the Fiji/ImageJ software. At the beginning of image analysis, image drifting was corrected using microtubule channels with an ImageJ plugin StackReg[46]. To reduce background signals of each images, custom Walking Average macro and the ImageJ subtract background tool were used.

In Figs. 1 and 2, GFP-labeled particles of Msd1, Wdr8, and MZT1a were first picked up and microtubule dynamics in the region of interest were then analyzed in the mCherry channel. To measure the time required to release γTuRC from microtubules (Fig. 2g), those MZT1a particles which appeared on the cortical microtubules and later disappeared after branch-forming nucleation during the image capture period (up to 600 s) were analyzed.

The dynamics of mCherry-TUB6-labeled microtubules was used to identify the nucleation events in which both initiation and release of daughter microtubules were visible (Fig. 3), and the crossover events in which both crossover formation and subsequent severing of overlaid microtubules were visible (Fig. 6). The signal intensities of GFP-KTN1 at the severed regions (5 pixel × 5 pixel; 0.52 μm²) were recorded over time.

Arrival to and disappearance from the cell cortex of Msd1a-mCherry and MZT1a-GFP were monitored by using Fiji/ImageJ plugin TrackMate[47] with the LoG detector filter and the detection spot radius size of 2.5 pixel (0.26 μm). Fluorescent particles were tracked by using the nearest-neighbor search filter with a maximal distance allowance setting of 5 pixel. The time lag (800 ms) required for the sequential excitations was corrected.

**Yeast two-hybrid interaction assay.** Open reading frames for *Msd1a*, *Msd1b*, and *Wdr8* cDNAs were obtained after reverse-transcription PCR from seedling RNA using primer sets in Supplementary Table S1, followed by cloning of the PCR fragments into pHSG399. The cDNAs were excised by BamHI and were inserted to the BamHI site in pGADT7 (Clontech) and pGBKT7 (Clontech) for translational fusions to the Gal4 activation domain and to the Gal4-binging domain, respectively. Other plasmids and experimental procedures have been previously reported[42].

**Phenotypic analysis.** Primary roots of *Arabidopsis* seedlings grown on vertically placed agar plates were photographed every day after 5–10 days of germination with a conventional image scanner (EPSON GT-S650) and the root length was analyzed by a FIJI/ImageJ software. To analyze cell shapes, primary roots of light-grown 4-day-old seedlings were stained with propidium iodide at 10 μg mL$^{-1}$ and observed with a confocal microscope Leica SP8 with a ×20, 0.75 NA HC PL APO CS2 objective (Leica). To analyze root cortex cells of the differentiation region, light-grown 4-day-old seedlings were fixed with 4% formaldehyde for 1 h, followed by several washes with the 1× phosphate-buffered saline, and then stained with a 1 : 5 diluted solution of Calcofluor White Stain (Sigma-Aldrich, #18909) for 30 min. After washing out an excess staining solution, root samples were mounted with the ClearSee solution for deep tissue imaging[48]. Images were taken as 0.7 μm-step z-stacks with a Zeiss inverted microscope Axio Observer equipped with a confocal system LSM800 and a ×40, 1.1 NA LD C-Apochromat water-immersion objective (Zeiss). The root differentiation zone was determined by the proto-xylem differentiation

**Statistical information.** Statistical analyses were performed using Microsoft Excel plugin software, StatPlus (AnalystSoft), Prism8 (GraphPad Software), and R software[49]. The data are presented as mean ± SD and *n*-values are shown in the figure legends. For boxplots, the bottom and top of each box represent the first and third quartiles, respectively. The horizontal line inside each box is the median (second quartile) and the red horizontal line represents the average. The whiskers range between 2nd percentile and 98th percentile. Data points outside this range are considered as outliers and are indicated as circles. n.s. indicates not significant,

$*p < 0.01$, $**p < 0.001$, $***p < 0.0001$. To compare means for multiple groups, Steel–Dwass test was performed using R software.

**Reporting summary.** Further information on research design is available in the Nature Research Reporting Summary linked to this article.

## Data availability
The mass spectrometry proteomics data have been deposited to the ProteomeXchange Consortium via the PRIDE [http://www.ebi.ac.uk/pride] partner repository with the dataset identifier PXD026063 and 10.6019/PXD026063. Other datasets generated during the current study are available from the corresponding authors on request. Source data are provided with this paper.

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

## Acknowledgements

We thank M. Kawaguchi for the help in the yeast two-hybrid assays and Y. Sakumura for discussion on data processing. This work was supported by Grant-in-Aid for Scientific Research (B) to T.H. (17H03698) and S.M. (15K21750, 15H05962, and 15H05955), and Human Frontier Science Program Grant to M.N. (18KK0195) from Japan Society for the Promotion of Science (JSPS), Japan, and NSF Grant (1158372) to D.W.E. ITbM was supported by World Premier International Research Center Initiative (WPI), Japan.

## Author contributions

T.H. and M.N. conceived the project. N.Y., M.N. and T.K. performed the experiments. T.H., M.N., N.Y. and D.W.E. analyzed the data and wrote the manuscript. All authors discussed the results and commented on the manuscript.

## Competing interests

The authors declare no competing interests.
