## [Peer Review File · Nature Communications]

REVIEWER COMMENTS

Reviewer #1 (Remarks to the Author):

Interphase plant cells possess highly dynamic microtubules in the cell cortex that undergo rapid reorganization, a phenomenon not found in animal cells. A key enzyme that drives microtubule remodeling is the severing protein katanin. An outstanding question is how katanin is targeted to branched microtubule nucleation sites in order to sever and release the newly born microtubules that frequently takes place in actively expanding plant cells. Results summarized in this manuscript revealed that the Msd1-Wdr8 complex functions in the recruitment of katanin to the nucleation site bearing the γ -tubulin complex in interphase but not mitotic cells. Therefore, they provided a mechanistic understanding of a key event taking place on cortical microtubules in plant cells.

Besides its significance specific to cortical microtubules, the work also demonstrated how evolutionarily conserved proteins are wired to regulate microtubule-severing events. It combined the beautiful Arabidopsis genetics and sophisticated live-cell imaging techniques that perhaps were often thought to be reserved for unicellular yeasts in the past. Therefore, the findings represent a significant breakthrough in understanding spatial regulation of microtubule dynamics in plant cells. Both the depth and quality are impressive so that the study is conclusive and informative.

The authors are invited to address a few points listed here:

1. Aside from the association between Msd1 and Wdr8, it was revealed that Msd1a but not Msd1b weakly interacted with GCP4 in a yeast two-hybrid assay. This finding led to two questions. Msd1a and 1b exhibit such a high sequence identity that one may ask how they could show a difference in the GCP interaction. Msd1 homolog in fission yeast interacts with Alp4/GCP2 with a defined interaction domain as reported in the cited work by Toya et al. (2007). Perhaps it is necessary to discuss the discrepancy here in Arabidopsis.
2. In figure 1a, Msd1-GFP purification rendered a higher yield than Wdr8-GFP. There seemed to be some "unidentified" bands. Were they degradation products of the mentioned proteins? More importantly, were any GCP proteins or katanin subunits co-purified with Msd1a?
3. The image data showed stronger relationship between Msd1-Wdr8 with katanin than with the γ -tubulin complex. Yet, the interaction assay was carried out between Msd1-Wdr8 and the γ -tubulin complex. One may question why it was not tested whether the complex directly interacts with katanin. This would provide an informative message because the functional relationship is a novel one.
4. It is intriguing that the Msd1-Wdr8 complex was recruited to meet with katanin at the branched nucleation site but not the crossover point. Does this suggest that the recruitment may require the γ -tubulin complex? The hypothesis is consistent with the finding that Msd1 arrives after the MZT1 appearance.
5. In the absence of Msd1-Wdr8 or katanin, branched microtubules underwent predominantly shrinkage. In the control cells, the category of "release" was dominant. A question here is whether these released microtubules also shrunk at the plus end. Obviously, the example shown in Figure 2d was a long branched microtubule. It would be interesting to learn whether shorter microtubules also were released in the control cells.
6. An intriguing phenomenon is that wdr8 suppressed ktn1 in terms of both microtubule release and seedling growth. It was interpreted that in the absence of katanin, Msd1-Wdr8 could stabilize the association of the γ -tubulin complex with the microtubule lattice. This model would be plausible if the Msd1-Wdr8 complex binds to both microtubules and the γ -tubulin complex. It was indicated that plant cells may share a similar mechanism of having a microtubule motor deliver the complex to meet with the γ -tubulin complex. However, the yeast work indicated that the motor activity was not required for Msd1-Wdr8 localization to the spindle pole body. The Figure 8 diagram would suggest a plus end-directed motor for the proposed action. The authors may want to reword the discussion regarding Msd1-Wdr8 association with the γ -tubulin complex.
7. In the sentence of lines 70-72, perhaps "mitosis" could be reworded to "cell division" in order to avoid confusion.
8. There was a sudden jump to the paragraph starting line 76. As written, readers may wonder why Msd1-Wdr8 was chosen for the study of microtubule-severing because the earlier description had only described its connection with microtubule nucleation.

Reviewer #2 (Remarks to the Author):

Microtubule nucleation and severing are the central events during microtubule organization, especially in accentrosomal plant cells. The manuscript entitled "An anchoring complex recruits katanin for microtubule severing at the plant cortical nucleation sites" by Yagi et al., uncovered that the conserved Msd1-Wdr8 complex has two antagonistic functions in Arabidopsis: stabilizing branching nucleation structures and recruiting katanin to sever the daughter microtubule. This work would provide important information and novel insights into the regulation of microtubule nucleation in plant cells, especially for the recruitment of katanin complexes at nucleation sites and the fate of daughter microtubules. The manuscript was well organized and clearly written, and the figures are of high quality. I only have some concerns over the biological significance of the Msd1-Wdr8 complex in plants, and I also believe that a number of points should be discussed better.

1. The authors claimed that Msd1 and Wdr8 form a heterodimeric complex (Line 126)? However, the data could not fully support this conclusion. In Arabidopsis, it is possible that all of these three components are required to form a functional complex, since all of these proteins was identified by the IP-LC/MS assay. At least, the authors should provide evidence showing the presence of Msd1a at nucleation sites of *msd1b* mutant cells, *vice versa*.

2. The authors stated that "Arabidopsis Msd1 binds microtubules and recruits Wdr8 to the microtubule lattice as a heteromeric protein complex" (line 104-105). However, in Fig. 1c, the Msd1a-GFP did not localize at MT nucleation site in the *wdr8* mutants, implying that Wdr8 is likely required for Msd1 recruitment. In addition, the localization of Msd1b-GFP in *wdr8* background is needed.

3. In Fig. 1f, the co-localized proportion for Msd1/Wdr8 and MZT1 needs to be evaluated with statistics. Prior work has shown that a considerable portion of the γ -TURC and the aumgmin complex are recruited to microtubule crossover sites (Wang et al., 2018, Current Biology). It is intriguing, but surprising why the conserved Msd1/Wdr8 complex, which is a centrosomal microtubule anchor complex, is only recruited at nucleation sites through interaction with GCP4 of the γ -TURC. There exist parallel nucleation and nucleation at crossover sites. Whether Msd1/Wdr8 is recruited at those sites as well? It must be presented explicitly and discussed.

4. Since the Msd1-Wdr8 complex affects the stability of γ -TURC, do they have effects on the nucleation events? It will be better to test the nucleation frequency, nucleation angle, and the fraction of branched and parallel form of nucleation events in *msd1a* *msd1b* mutants and *wdr8* mutants.

5. What is the evidence that the Msd1-Wdr8 complex functions to stabilize branching nucleation structures? However, it is consuming that, in Fig. 2b and 2g, the mean life time of MZT1-GFP and the mean release time were increased in the *msd1a* *msd1b* cells and the *wdr8* cells.

6. The genetic evidence showing that *msd1/wdr8* mutation could partially rescue the defects of the *ktn1* mutant is quite intriguing. However, the interpretation is confusing. It must be presented explicitly and discussed. Actually, the aspect is very important, but the title of this manuscript only mentioned the first function.

7. No evidence supports katanin may interact with γ -TURC. In Fig 6, the model should be revised.

8. There are some minor errors with the grammar and format. For example: Page 6 Line 123 and 124, *Wrd8* should be *Wdr8*. Page 10 Line 219 and 220 should be on the same line.

Responses to Reviewers' comments

First of all, we would like to thank two anonymous reviewers for critical reading of our manuscript and many valuable comments. Based on these comments, we conducted several experiments and analyses, and revised the manuscript. For re-submission, we also formatted the manuscript in accordance with the manuscript guideline of Nature Communications. These changes include reduction of abstract length and creation of subsections in the Results section. For the two subsections, we moved two figures from Supplementary materials to the main figures so that Figs. 5 and 6 now appear as main figures. New results obtained are shown Fig. 3f,g., Fig. S3, Fig. S4, Fig. S6, and six new movies (Videos 9, 12, 13, 14, 15 and 16). Numbering of figures were updated accordingly. Changes are indicated in red in the revised manuscript. We hope that these revisions are satisfactory and our revised manuscript will be accepted for publication.

Reviewer #1 (Remarks to the Author):

1. Aside from the association between Msd1 and Wdr8, it was revealed that Msd1a but not Msd1b weakly interacted with GCP4 in a yeast two-hybrid assay. This finding led to two questions. Msd1a and 1b exhibit such a high sequence identity that one may ask how they could show a difference in the GCP interaction. Msd1 homolog in fission yeast interacts with Alp4/GCP2 with a defined interaction domain as reported in the cited work by Toya et al. (2007). Perhaps it is necessary to discuss the discrepancy here in Arabidopsis.

We repeated Y2H assays several times over 2-3 years using independently isolated yeast colonies, and found that Msd1a and Msd1b interacted robustly with Wdr8, and sometimes showed weak interactions with GCP2 and GCP4 (but never with other components of gTuRC). Their weak interactions with GCP2 and GCP4 are not always reproducible; for example, in one experiment, one Msd1 isoform interacted with both GCPs and in other experiments both Msd1s showed positive results with either GCPs. Since we are currently unable to set up experimental conditions that enable us to demonstrate consistent and reproducible interaction results, we decided to remove these results from the main text and Supplementary figures. This revision, we believe, will not undermine main conclusions in the paper. The reported interaction between Msd1 and GCP2 (Alp4) in fission yeast was mentioned in the revised text.

2. In figure 1a, Msd1-GFP purification rendered a higher yield than Wdr8-GFP. There seemed to be some “unidentified” bands. Were they degradation products of the mentioned proteins? More importantly, were any GCP proteins or katanin subunits co-purified with Msd1a?

We analyzed some of these unidentified bands. They are not GCP proteins nor katanin. However, a particular kinesin was identified in both Msd1 and Wdr8 precipitates. We are currently working on this kinesin to see whether it transports Msd1-Wdr8 to the cortical nucleation sites. Hopefully we will report on this in our next paper.

3. The image data showed stronger relationship between Msd1-Wdr8 with katanin than with the γ -tubulin complex. Yet, the interaction assay was carried out between Msd1-Wdr8 and the γ -tubulin complex. One may question why it was not tested whether the complex directly interacts with katanin. This would provide an informative message because the functional relationship is a novel one.

Yes, we have examined whether Msd1 and Wdr8 interact with p60 and p80 subunits of katanin in Y2H assays, but did not detect any interactions. Possibly, their interactions are transient or require other proteins.

4. It is intriguing that the Msd1-Wdr8 complex was recruited to meet with katanin at the branched nucleation site but not the crossover point. Does this suggest that the recruitment may require the γ -tubulin complex? The hypothesis is consistent with the finding that Msd1 arrives after the MZT1 appearance.

The Msd1-Wdr8 complex is not recruited to the crossover sites, possibly because gTuRC is not present there. Our study shows that recruitment of katanin to the cortical nucleation sites, which contain gTuRC, requires the Msd1-Wdr8 complex, but we are still open to the question whether katanin directly interacts with Msd1, Wdr8, or any components of gTuRC. Future studies will hopefully address this point.

5. In the absence of Msd1-Wdr8 or katanin, branched microtubules underwent predominantly shrinkage. In the control cells, the category of “release” was dominant. A question here is whether these released microtubules also shrunk at the plus end. Obviously, the example shown in Figure 2d was a long branched microtubule. It would be interesting to learn whether shorter microtubules also were released in the control

cells.

The released daughter microtubules migrate away from the original nucleation sites by net polymerization (with dynamic instability) at the plus end and rather slow and consistent depolymerization at the minus end. This hybrid treadmilling behavior of released daughter microtubules is essentially the same with those free cortical microtubules reported earlier (e.g., Shaw et al., *Science*, 2003). We did not examine microtubule release events in relation to the length of daughter microtubules. Although this is a potentially interesting question, other studies may address this in the future.

6. An intriguing phenomenon is that *wdr8* suppressed *ktn1* in terms of both microtubule release and seedling growth. It was interpreted that in the absence of katanin, Msd1-Wdr8 could stabilize the association of the γ -tubulin complex with the microtubule lattice. This model would be plausible if the Msd1-Wdr8 complex binds to both microtubules and the γ -tubulin complex. It was indicated that plant cells may share a similar mechanism of having a microtubule motor deliver the complex to meet with the γ -tubulin complex. However, the yeast work indicated that the motor activity was not required for Msd1-Wdr8 localization to the spindle pole body. The Figure 8 diagram would suggest a plus end-directed motor for the proposed action. The authors may want to reword the discussion regarding Msd1-Wdr8 association with the γ -tubulin complex.

In fission yeast, a minus-end directed kinesin Pkl1 is shown to transport Msd1-Wdr8 through the spindle microtubule toward the mitotic SPB (Yukawa et al., *JCB* 2015). In that paper, a ternary Msd1-Wdr8 complex (labeled with GFP) containing a non-motile Pkl1 rigor mutant was artificially tethered to the SPB by using an Alp4 form fused to GFP-binding protein (GBP), to test the role of Pkl1 motor activity on the microtubule protrusion phenotype. This particular experiment does not demonstrate that the motor activity is not required for Msd1-Wdr8 localization to the SPB. We modified the model to represent the Msd1-Wdr8 complex translocates toward the minus end of the mother microtubule (although the microtubule polarity is not explicitly shown for simplicity).

7. In the sentence of lines 70-72, perhaps “mitosis” could be reworded to “cell division” in order to avoid confusion.

Revised as suggested.

8. There was a sudden jump to the paragraph starting line 76. As written, readers may wonder why Msd1-Wdr8 was chosen for the study of microtubule-severing because the earlier description had only described its connection with microtubule nucleation.

We inserted a sentence describing paper conclusions at the end of the introductory part to improve the flow of thoughts.

Reviewer #2 (Remarks to the Author):

1. The authors claimed that Msd1 and Wdr8 form a heterodimeric complex (Line 126)? However, the data could not fully support this conclusion. In Arabidopsis, it is possible that all of these three components are required to form a functional complex, since all of these proteins was identified by the IP-LC/MS assay. At least, the authors should provide evidence showing the presence of Msd1a at nucleation sites of *msd1b* mutant cells, vice versa.

Arabidopsis lines expressing Msd1a-GFP and Msd1b-GFP are now being crossed respectively to *msd1b* and *msd1a* mutants, but it will take several months to obtain suitable plant lines for observation. Because Msd1a and Msd1b show high amino acid sequence identity, we assumed that they function redundantly and did not examine single mutant phenotypes in this study.

2. The authors stated that “Arabidopsis Msd1 binds microtubules and recruits Wdr8 to the microtubule lattice as a heteromeric protein complex” (line 104-105). However, in Fig. 1c, the Msd1a-GFP did not localize at MT nucleation site in the *wdr8* mutants, implying that Wdr8 is likely required for Msd1 recruitment. In addition, the localization of Msd1b-GFP in *wdr8* background is needed.

Since we did not generate Msd1b-GFP marker lines in the *wdr8* background, it is currently not possible to analyze them. Again, there is no indication so far that Msd1a and Msd1b have distinct functions. We hope the reviewer to understand that it is not practical for a small lab to duplicate all the Msd1a data with Msd1b.

3. In Fig. 1f, the co-localized proportion for Msd1/Wdr8 and MZT1 needs to be evaluated

with statistics. Prior work has shown that a considerable portion of the γ -TURC and the aumgmin complex are recruited to microtubule crossover sites (Wang et al., 2018, Current Biology). It is intriguing, but surprising why the conserved Msd1/Wdr8 complex, which is a centrosomal microtubule anchor complex, is only recruited at nucleation sites through interaction with GCP4 of the γ -TURC. There exist parallel nucleation and nucleation at crossover sites. Whether Msd1/Wdr8 is recruited at those sites as well? It must be presented explicitly and discussed.

Sub-cellular localization of Msd1a, Msd1b, and Wdr8 are summarized in Fig. 1e, with a reference to MZT1. Time-lapse analysis of cortical Msd1a and MZT1 appearance is shown in Fig. 1g. We hope that these data largely address the comment raised by the reviewer.

The Msd1-Wdr8 complex is not generally recruited to the crossover sites, which supports that katanin localization at crossover sites is not affected by the wdr8 mutation. Relevant results are shown in Fig. 6 (now moved from Supplementary files) and Videos 23 and 24, and are described in page 13.

Based on the reviewer's comment, we examined whether Msd1-Wdr8 is involved in katanin recruitment in bundle-forming parallel nucleation. In wdr8 cells, GFP-KTN1 particles are not recruited to the nucleation sites of parallel daughter microtubules, indicating that both branching and bundle-forming nucleation types require Wdr8 for katanin recruitment. The results are shown in Fig. 3f,g and described in page 10.

4. Since the Msd1-Wdr8 complex affects the stability of γ -TURC, do they have effects on the nucleation events? It will be better to test the nucleation frequency, nucleation angle, and the fraction of branched and parallel form of nucleation events in msd1a msd1b mutants and wdr8 mutants.

Based on the reviewer's comment, we analyzed these characteristic related to the nucleation events. The results are shown in Supplementary Fig. S6 and described in pages 11 and 12. In the wdr8 cells, relative fractions of branched and parallel forms are somewhat affected but other events are not.

5. What is the evidence that the Msd1-Wdr8 complex functions to stabilize branching nucleation structures? However, it is consuming that, in Fig. 2b and 2g, the mean life

time of MZT1-GFP and the mean release time were increased in the *msd1a msd1b* cells and the *wdr8* cells.

In *ktn1* single mutant cells, the nucleation sites are stable and are seldom released. However, in the *ktn1 wdr8* double mutant cells, the daughter microtubules are frequently released, although with some delay compared to the katanin-dependent release in wild-type cells.

6. The genetic evidence showing that *msd1/wdr8* mutation could partially rescue the defects of the *ktn1* mutant is quite intriguing. However, the interpretation is confusing. It must be presented explicitly and discussed. Actually, the aspect is very important, but the title of this manuscript only mentioned the first function.

The *msd1/wdr8* mutations substantially rescue anisotropic cell expansion (interphase) phenotypes of *ktn1* plants but not mitotic defects. This observation is consistent with the cellular phenotypes that *Msd1-Wdr8* is required for katanin recruitment to interphase cortical nucleation sites but not to mitotic nucleation sites.

Since the journal guideline required an abstract length less than approximately 150 words and since the original abstract already contains 227 words, we revised and substantially shortened the abstract while incorporating what the reviewer suggested (now the abstract contains 149 words). The article title should be less than 15 words (currently 14 words) and is difficult to accommodate more information as suggested. We expect that readers will get sufficient information from the revised abstract (and the text).

7. No evidence supports katanin may interact with γ -TuRC. In Fig 6, the model should be revised.

We do not have experimental evidence that katanin directly interacts with *Msd1*, *Wdr8* or component of γ TuRC, but have shown that *Msd1-Wdr8* is critical for katanin recruitment to the γ TuRC-containing cortical nucleation sites. In the legend of Fig. 6, we explicitly described that direct interaction between *Msd1-Wdr8* and katanin is yet to be shown.

8. There are some minor errors with the grammar and format. For example: Page 6 Line

123 and 124, Wrd8 should be Wdr8. Page 10 Line 219 and 220 should be on the same line.

Corrected.

REVIEWER COMMENTS

Reviewer #1 (Remarks to the Author):

Among the questions raised previously, the most important one was about the triangular relationship among the gamma-tubulin complex, the katanin complex, and this newly characterized Msd1-Wdr8 complex. It is recognized that their interactions are likely transient and difficult to be captured by protein purification. Fortunately, the authors have provided ample evidence of both live-cell imaging and genetic interactions among the mutations. Therefore, the manuscript presents a conclusive story and brings significant mechanistic insights into the spatial regulation of microtubule severing for the establishment of a dynamic cortical microtubule array.

In the response letter and discussion section, the authors indicated that their protein purification data resulted in recovering a kinesin motor. It is agreed that further characterization of the motor and its relationship with proteins included in this story would form another exciting story.

Reviewer #2 (Remarks to the Author):

This manuscript revealed the roles of Msd1-Wdr8 complex both in recruiting KTN1 at MT nucleation sites and in stabilizing g-TURC as well. Although this revised version has been improved, some serious concerns remain to be addressed before this paper could be accepted for publication.

In this paper, the authors claimed that the Msd1-Wdr8 complex functions to recruit katanin to the MT nucleation sites but not to MT crossovers. The authors should show the overall localization pattern of the Msd1-Wdr8 complex on MTs, and detail localization changes of one component in the mutant background of another component. For, example, based on current data, the authors assumed that Msd1a and Msd1b play redundant roles, and numerous analyses were established on this assumption. However, the weakest part of this manuscript is lack of direct evidence using the msd1a msd1b double mutant. In addition, this reviewer still thinks that the authors should observe Msd1b-GFP behaviors in the wdr8 background. Without solid genetic evidence, the conclusion could not be convincing.

The author concluded that each puncta signal of Msd1 or Wdr8 represents the Msd1-Wdr8 complex, thus there was no detectable signals of Msd1 in wdr8 background and no Wdr8 signal at MT nucleation sites in the msd1a/msd1b double mutant, respectively (Fig 1c and 1d). Given that a considerable amount of Msd1 or Wdr8 are localized on MTs where no nucleation occurs, images showing multiple patterns of Msd1 localization in wdr8 background are required, instead of illustrating an individual nucleation event, vice-versa. It will be more confidential to conclude that functional Msd1-Wdr8 complexes are formed on MTs if Msd1 can not be detected on the MTs of wdr8 mutants.

Although Msd1a, Msd1b, and Wdr8 particles were observed as stable puncta along MTs with a similar distribution pattern to that of MZT1 (Fig 1e). It remains unclear the percentage of Msd1/Wdr8 puncta that are co-localized with MZT1, which would be important to evaluate the Msd1/Wdr8 localization on MTs. The author claimed that it is likely the Msd1 that recruits Wdr8 to form a functional complex on MTs (Fig 1b). However, it is difficult to see the cytosolic pattern. Furthermore, if Msd1 acts as the recruit factor of Wdr8, the localization of Msd1 should be unaffected in wdr8 mutants and Wdr8 should be abolished on MTs in the msd1a/msd1b double mutant.

The authors find the second function of Msd1-Wdr8 is to stabilize the g-TURC on MTs. The release frequency of daughter MTs was decreased in wdr8 or msd1a/msd1b mutants (Fig. 2f). However, in the wdr8 mutants, the duration time of MZT1 was prolonged (Fig. 2g). This is contradictory with the notion that Wdr8 stabilizes the nucleation structure via association with g-TURC.

Responses to the reviewers' comments

Reviewer #1 (Remarks to the Author):

Among the questions raised previously, the most important one was about the triangular relationship among the gamma-tubulin complex, the katanin complex, and this newly characterized Msd1-Wdr8 complex. It is recognized that their interactions are likely transient and difficult to be captured by protein purification. Fortunately, the authors have provided ample evidence of both live-cell imaging and genetic interactions among the mutations. Therefore, the manuscript presents a conclusive story and brings significant mechanistic insights into the spatial regulation of microtubule severing for the establishment of a dynamic cortical microtubule array. In the response letter and discussion section, the authors indicated that their protein purification data resulted in recovering a kinesin motor. It is agreed that further characterization of the motor and its relationship with proteins included in this story would form another exciting story.

Answer:

Thank you for understanding technical challenges to address fragile physical interactions among the Msd1-Wdr8 complex, the gamma-tubulin complex, and the katanin complex. In the near future, we hope to publish on the putative kinesin once its role related to the Msd1-Wdr8 complex is sufficiently characterized.

Reviewer #2 (Remarks to the Author):

1. In this paper, the authors claimed that the Msd1-Wdr8 complex functions to recruit katanin to the MT nucleation sites but not to MT crossovers. The authors should show the overall localization pattern of the Msd1-Wdr8 complex on MTs, and detail localization changes of one component in the mutant background of another component. For, example, based on current data, the authors assumed that Msd1a and Msd1b play redundant roles, and numerous analyses were established on this assumption. However, the weakest part of this manuscript is lack of direct evidence using the msd1a msd1b double mutant. In addition, this reviewer still thinks that the authors should observe Msd1b-GFP behaviors in the wdr8 background. Without solid genetic evidence, the conclusion could not be convincing.

Answer:

Msd1a and Msd1b proteins show high homology, and their encoding genes are expressed in highly overlapping cell types (the information was added in the revised manuscript). Co-purification of both proteins, their Y2H interactions with Wdr8, and identical subcellular localization all indicate that they likely have redundant functions. Accordingly, we do not think that analyses of Msd1b-GFP localization in the *wdr8* mutant background is an essential experiment (as explained in our previous response letter, such materials are currently not available). Even without this proposed experiment, the conclusions in this paper are solid and do not change.

It is difficult to understand why this reviewer thinks that providing further evidence for the expected genetic redundancy is essential for this paper. We are not proposing that these two Msd1 genes have non-overlapping functions. Surely, to propose their distinct functions (which are unlikely), further experiments and detailed analyses are necessary. If they do have partly distinct functions in certain cellular activities or in response to environmental stimuli, these results will be better covered by a new independent publication, but not in this paper.

2. The author concluded that each puncta signal of Msd1 or Wdr8 represents the Msd1-Wdr8 complex, thus there was no detectable signals of Msd1 in *wdr8* background and no Wdr8 signal at MT nucleation sites in the *msd1a/msd1b* double mutant, respectively (Fig 1c and 1d). Given that a considerable amount of Msd1 or Wdr8 are localized on MTs where no nucleation occurs, images showing multiple patterns of Msd1 localization in *wdr8* background are required, instead of illustrating an individual nucleation event, vice-versa. It will be more confidential to conclude that functional Msd1-Wdr8 complexes are formed on MTs if Msd1 can not be detected on the MTs of *wdr8* mutants.

Answer:

We included lower magnification images that cover larger cortical area of wild-type, *msd1* double mutant, and *wdr8* cells, in which microtubules and Msd1 or Wdr8 are labeled (in the new Supplementary Figure S3). These images address to the comment, and support our conclusions.

3. Although Msd1a, Msd1b, and Wdr8 particles were observed as stable puncta along MTs with a similar distribution pattern to that of MZT1 (Fig 1e). It remains unclear the percentage of Msd1/Wdr8 puncta that are co-localized with MZT1, which would be important to evaluate the Msd1/Wdr8 localization on MTs. The author claimed that it is

likely the Msd1 that recruits Wdr8 to form a functional complex on MTs (Fig 1b). However, it is difficult to see the cytosolic pattern. Furthermore, if Msd1 acts as the recruit factor of Wdr8, the localization of Msd1 should be un-affected in *wdr8* mutants and Wdr8 should be abolished on MTs in the *msd1a/msd1b* double mutant.

Answer:

In the new Figure 1f, we have now included co-localization images of MZT1 and Msd1a particles that stayed on the cell cortex for longer than 10 s, and presented (in the revised text) percentages of their co-localization. Most of the two particles do overlap. Since particles in the cytoplasm rapidly move and do not stay in fixed locations, they are not visible as distinct puncta, and abundant cytoplasmic localization increases the background fluorescence in the cytoplasm. Punctate localization patterns of Msd1 and Wdr8 require formation of a heteromeric complex; in the absence of their interacting partner such localization patterns are lost (which is what we observed).

4. The authors find the second function of Msd1-Wdr8 is to stabilize the γ -TURC on MTs. The release frequency of daughter MTs was decreased in *wdr8* or *msd1a/msd1b* mutants (Fig. 2f). However, in the *wdr8* mutants, the duration time of MZT1 was prolonged (Fig. 2g). This is contradictory with the notion that Wdr8 stabilizes the nucleation structure via association with γ -TURC.

Answer:

Here the reviewer is apparently misunderstood. Possibly, the use of the word, “stabilize” in some sentences may be partly responsible for such misunderstanding. To avoid/minimize confusion and misunderstanding, we rephrased and changed the words “stabilize” or “stabilization” to “persist” or “persistence” in the relevant sentences in the revised text. Msd1-Wdr8 has dual functions; it stabilizes cortical nucleation structures, and recruits katanin to sever the structures. Our results are consistent with these proposed roles.

1. In this paper, the authors claimed that the Msd1-Wdr8 complex functions to recruit katanin to the MT nucleation sites but not to MT crossovers. The authors should show the overall localization pattern of the Msd1-Wdr8 complex on MTs, and detail localization changes of one component in the mutant background of another component. For, example, based on current data, the authors assumed that Msd1a and Msd1b play redundant roles, and numerous analyses were established on this assumption. However, the weakest part of this manuscript is lack of direct evidence using the *msd1a msd1b* double mutant. In addition, this reviewer still thinks that the authors should observe Msd1b-GFP behaviors in the *wdr8* background. Without solid genetic evidence, the conclusion could not be convincing.

Answer:

Msd1a and Msd1b proteins show high homology, and their encoding genes are expressed in highly overlapping cell types (the information was added in the revised manuscript). Co-purification of both proteins, their Y2H interactions with Wdr8, and identical subcellular localization all indicate that they likely have redundant functions. Accordingly, we do not think that analyses of Msd1b-GFP localization in the *wdr8* mutant background is an essential experiment (as explained in our previous response letter, such materials are currently not available). Even without this proposed experiment, the conclusions in this paper are solid and do not change.

It is difficult to understand why this reviewer thinks that providing further evidence for the expected genetic redundancy is essential for this paper. We are not proposing that these two Msd1 genes have non-overlapping functions. Surely, to propose their distinct functions (which are unlikely), further experiments and detailed analyses are necessary. If they do have partly distinct functions in certain cellular activities or in response to environmental stimuli, these results will be better covered by a new independent publication, but not in this paper.

This reviewer's comments: Although this reviewer still thinks that supplement of a simple experiment deserves publication at the high-profile journal *Nature Communications*, I understand the unavailability of the materials currently. My only concern is why the growth phenotype of the *wdr8* mutant and the *msd1a msd1b* double mutant is indistinguishable with the wild type since the Msd1-Wdr8 complex is assumed to play so important roles. The authors need to discuss this point.

2. The author concluded that each puncta signal of Msd1 or Wdr8 represents the Msd1-Wdr8 complex, thus there was no detectable signals of Msd1 in *wdr8* background and no Wdr8 signal at MT nucleation sites in the *msd1a/msd1b* double mutant, respectively (Fig 1c and 1d). Given that a considerable amount of Msd1 or Wdr8 are localized on MTs where no nucleation occurs, images showing multiple patterns of Msd1 localization in *wdr8* background are required, instead of illustrating an individual nucleation event, vice-versa. It will be more confidential to conclude that functional

Msd1-Wdr8 complexes are formed on MTs if Msd1 can not be detected on the MTs of wdr8 mutants.

Answer:

We included lower magnification images that cover larger cortical area of wild-type, msd1 double mutant, and wdr8 cells, in which microtubules and Msd1 or Wdr8 are labeled (in the new Supplementary Figure S3). These images address to the comment, and support our conclusions.

This reviewer's comments: It is weird for the authors to use a single average projection of 151 image frames (302s) to illustrate the localization of Msd1 or Wdr8 because these particles do NOT persistently localize at fixed sites of MTs for long time. Flashing fluorescent signals like Msd1 or Wdr8 would disappear under average projection processing. The authors should provide single-frame images or original image series.

3. Although Msd1a, Msd1b, and Wdr8 particles were observed as stable puncta along MTs with a similar distribution pattern to that of MZT1 (Fig 1e). It remains unclear the percentage of Msd1/Wdr8 puncta that are co-localized with MZT1, which would be

important to evaluate the Msd1/Wdr8 localization on MTs. The author claimed that it is likely the Msd1 that recruits Wdr8 to form a functional complex on MTs (Fig 1b). However, it is difficult to see the cytosolic pattern. Furthermore, if Msd1 acts as the recruit factor of Wdr8, the localization of Msd1 should be un-affected in wdr8 mutants and Wdr8 should be abolished on MTs in the msd1a/msd1b double mutant.

Answer:

In the new Figure 1f, we have now included co-localization images of MZT1 and Msd1a particles that stayed on the cell cortex for longer than 10 s, and presented (in the revised text) percentages of their co-localization. Most of the two particles do overlap. Since particles in the cytoplasm rapidly move and do not stay in fixed locations, they are not visible as distinct puncta, and abundant cytoplasmic localization increases the background fluorescence in the cytoplasm. Punctate localization patterns of Msd1 and Wdr8 require formation of a heteromeric complex; in the absence of their interacting partner such localization patterns are lost (which is what we observed).

This reviewer's comments: Current evidence could not fully support the notion that Msd1 acts as a recruitment factor for Wdr8. The increase of background fluorescence may not represent a cytoplasm pattern of Wdr8 because different cells usually show different fluorescence background.

4. The authors find the second function of Msd1-Wdr8 is to stabilize the g-TURC on MTs. The release frequency of daughter MTs was decreased in wdr8 or msd1a/msd1b mutants (Fig. 2f). However, in the wdr8 mutants, the duration time of MZT1 was prolonged (Fig. 2g). This is contradictory with the notion that Wdr8 stabilizes the nucleation structure via association with g-TURC.

Answer:

Here the reviewer is apparently misunderstood. Possibly, the use of the word, “stabilize” in some sentences may be partly responsible for such misunderstanding. To avoid/minimize confusion and misunderstanding, we rephrased and changed the words “stabilize” or “stabilization” to “persist” or “persistence” in the relevant sentences in the revised text. Msd1-Wdr8 has dual functions; it stabilizes cortical nucleation structures, and recruits katanin to sever the structures. Our results are consistent with these proposed roles.

This reviewer’s comments: For example, MZT1 plays an important role to stabilize the nucleation structure. Longer duration time of MZT at MT nucleation sites results in more stable nucleation structure. Thus, we still think that the authors need to clarify this issue in the discussion.

1. Although this reviewer still thinks that supplement of a simple experiment deserves publication at the high-profile journal *Nature Communications*, I understand the unavailability of the materials currently. My only concern is why the growth phenotype of the *wdr8* mutant and the *msd1a msd1b* double mutant is indistinguishable with the wild type since the Msd1-Wdr8 complex is assumed to play so important roles. The authors need to discuss this point.

The counteracting functions of the Msd1-Wdr8 complex for sequentially stabilizing and de-stabilizing cortical nucleation sites likely underlie apparently indistinguishable growth of the mutants with the wild type under standard growth conditions. We explicitly added this explanatory sentence in the discussion part (in the middle of the second paragraph).

2. It is weird for the authors to use a single average projection of 151 image frames (302s) to illustrate the localization of Msd1 or Wdr8 because these particles do NOT persistently localize at fixed sites of MTs for long time. Flashing fluorescent signals like Msd1 or Wdr8 would disappear under average projection processing. The authors should provide single-frame images or original image series.

Majority of the Msd1/Wdr8 particle signals were not eliminated after stacked projection processing, but as the reviewer indicates some particles with short resident times became difficult to discern. To cope with this issue, we now prepared four new movie files in which Msd1 or Wdr8 particle dynamics are monitored in wild-type and mutant cells (Movies 5 to 9; subsequent movies were re-numbered). These additional movie data clearly show that stable cortical localization of Msd1 and Wdr8 requires each other.

3. Current evidence could not fully support the notion that Msd1 acts as a recruitment factor for Wdr8. The increase of background fluorescence may not represent a cytoplasm pattern of Wdr8 because different cells usually show different fluorescence background.

The reviewer appears to refer to the sentence, “These results are consistent with the hypothesis that Arabidopsis Msd1 binds microtubules and recruits Wdr8 to the microtubule lattice as a heteromeric protein complex” in page 5 of the previous text. Similar subcellular localization results of yeast Msd1 and Wdr8 supports our interpretation (J. Cell Biol. 209: 549-562, 2015). Nevertheless, since the description of

Msd1 as a recruitment factor of Wdr8 seems to annoy the reviewer, we now deleted this whole sentence (which is not essential for the flow of the text) in the new text.

4. For example, MZT1 plays an important role to stabilize the nucleation structure. Longer duration time of MZT at MT nucleation sites results in more stable nucleation structure. Thus, we still think that the authors need to clarify this issue in the discussion.

Now that high resolution structures and biochemical properties of several gTuRCs have been understood recently (see J. Cell Biol. 220, e202009146, 2021, Curr. Opin. Cell Biol. 68: 124-131, 2021, and references therein), it is established that MZT1 is localized in a luminal bridge of the complex and stabilizes the nucleation-competent form of gTuRCs. However, no evidence so far indicates that MZT1 stabilizes the association of the gTuRCs on the microtubule lattice or at the branching/parallel nucleation structures. Based on the available knowledge on microtubule nucleation, our interpretation of the results is straightforward and reasonable. Since reasonable alternative interpretations are available, our current discussions are solid.

REVIEWERS' COMMENTS

Reviewer #2 (Remarks to the Author):

Most of my concerns have been addressed. The manuscript can be accepted.